# Involvement of skin TRPV3 in temperature detection regulated by TMEM79 in mice

Jing Lei [1,2,3], Reiko U. Yoshimoto [4], Takeshi Matsui [5,6,7], Masayuki Amagai[5,7], Mizuho A. Kido [4] & Makoto Tominaga [1,2,3] ✉

TRPV3, a non-selective cation transient receptor potential (TRP) ion channel, is activated by warm temperatures. It is predominantly expressed in skin keratinocytes, and participates in various somatic processes. Previous studies have reported that thermosensation in mice lacking TRPV3 was impaired. Here, we identified a transmembrane protein, TMEM79, that acts as a negative regulator of TRPV3. Heterologous expression of TMEM79 was capable of suppressing TRPV3-mediated currents in HEK293T cells. In addition, TMEM79 modulated TRPV3 translocalization and promoted its degradation in the lysosomes. TRPV3-mediated currents and $Ca^{2+}$ influx were potentiated in primary mouse keratinocytes lacking TMEM79. Furthermore, TMEM79-deficient male mice preferred a higher temperature than did wild-type mice due to elevated TRPV3 function. Our study revealed unique interactions between TRPV3 and TMEM79, both in vitro and in vivo. These findings support roles for TMEM79 and TRPV3 in thermosensation.

Thermosensation is a crucial sensory process for survival in all organisms. In mammals, many thermoreceptors that respond to thermal stimuli have been identified, including transient receptor potential (TRP) channels[1]. TRP channels are non-selective cation ion channels. In humans, 27 TRP channels have been identified, and they have been divided into six subfamilies based on their amino acid sequences. Among them, there are 11 thermo-sensitive TRP channels so far, but only a few of them have been definitively demonstrated to be required for sensing either unpleasant cold or noxious heat in vivo, such as TRPV1, TRPA1, and TRPM8 that are mainly located in dorsal root ganglion (DRG) neurons[2–4]. However, the mechanism by which our bodies deliver warm temperature information through the skin is not entirely clarified. It is well-accepted that temperature-sensitive sensory neurons embedded in the skin could convey thermal cues from the periphery to the central nervous system (CNS). Importantly, non-neuronal cells like keratinocytes, the most abundant cells in the epidermis, were recently proven to be required for temperature detection[5]. In addition, it has been reported that TRPV3 and TRPV4 are important for warmth sensation because of their warmth-induced activation and prevalent expression in keratinocytes and because of the abnormal behavior of mutant mice in temperature gradient assays[6–9]. Our previous study reported that the activation of TRPV3 by temperature could transmit temperature information from skin keratinocytes to sensory neurons through the release of adenosine triphosphate (ATP)[10]. Therefore, it is important to understand whether the activities of TRPV3 or TRPV4 in keratinocytes directly or indirectly contact nerve endings in the skin.

In addition to thermosensitivity (threshold > 33 °C), TRPV3 can also be activated by some monoterpenes that are derived from natural plants, including camphor, menthol, carvacrol, and thymol, as well as synthetic chemicals like 2-APB[11–13]. TRPV3 is involved in numerous physiological and pathophysiological processes in the skin. For

[1]Division of Cell Signaling, National Institute for Physiological Sciences, National Institutes of Natural Sciences, 444-8787 Okazaki, Japan. [2]Department of Physiological Sciences, The Graduate University for Advanced Studies (SOKENDAI), 444-8585 Okazaki, Japan. [3]Thermal Biology Group, Exploratory Research Center on Life and Living Systems, National Institutes of Natural Sciences, 444-8787 Okazaki, Japan. [4]Division of Histology and Neuroanatomy, Department of Anatomy and Physiology, Faculty of Medicine, Saga University, 849-8501 Saga, Japan. [5]Laboratory for Skin Homeostasis, RIKEN Center for Integrative Medical Sciences, 230-0045 Yokohama, Japan. [6]Laboratory for Evolutionary Cell Biology of the Skin, School of Bioscience and Biotechnology, Tokyo University of Technology, 1404-1 Katakura, Hachioji 192-0982 Tokyo, Japan. [7]Department of Dermatology, Keio University School of Medicine, 160-8582 Tokyo, Japan. ✉e-mail: tominaga@nips.ac.jp

instance, TRPV3 contributes to skin barrier formation, wound healing, itch, and hair growth[14–21]. In addition, many TRPV3 gain-of-function mutations (G573S, G573C, G573A, G568V, R416Q, R416W, L655P, W692S, and L694P) have been found in human patients with *Olmsted syndrome* (OS), a hyperkeratotic skin disease[22,23]. In regard to TRPV3's function, TMEM79 is a putative five-pass transmembrane protein sharing similar dermal functions to TRPV3, including high expression in skin keratinocytes and involvement in itching or skin barrier formation. In this study, we found that TMEM79 reduced TRPV3-mediated activities via protein-protein interactions[24–28]. Moreover, TMEM79-knockout mice rapidly responded to warmer temperatures in a thermal gradient, which is likely attributed to the increased levels of TRPV3. This finding reveals a previously unreported modulation of TRPV3 ion channels and indicates that skin keratinocytes play a role in conveying thermal information.

## Results

### TMEM79 reduces currents mediated by TRPV3, but not those of TRPV4

We examined whether TMEM79 had the ability to regulate TRPV3 activity in a heterologous system. Thus, we performed electrophysiological experiments using a whole-cell patch–clamp method using HEK293T cells transiently expressing mouse TRPV3 (mTRPV3) alone or in combination with mouse TMEM79 (mTMEM79) (Fig. 1). Application of 300 μM 2-APB, a TRPV3 agonist, in cells expressing both mTRPV3 and mTMEM79 (Fig. 1a, b, red) resulted in smaller current responses than in cells expressing mTRPV3 alone without clear changes in voltage-dependence (Fig. 1a, b, black). We next measured the currents at different concentrations of 2-APB among the two expression patterns and established dose-dependence curves. Co-expression of TMEM79 reduced TRPV3 currents in a dose-dependent manner both at +60 (Fig. 1c) and −60 mV (Fig. 1d). Decreased TRPV3 currents could be attributed to either a change in dose-dependency for 2-APB or a reduction in the number of TRPV3 channels in the cells. Therefore, we evaluated the $EC_{50}$ values. Mean $EC_{50}$ values were $57.4 \pm 3.2$ and $57.8 \pm 20.8$ μM with and without TMEM79, respectively, at a holding potential of +60 mV. Similarly, mean $EC_{50}$ values were $88.3 \pm 13.6$ and $83.5 \pm 18.9$ μM with and without TMEM79, respectively, at −60 mV. These results indicate that the efficacy, but not the potency, was affected by the co-expression of TMEM79, suggesting that the differences in current amplitudes were likely caused by differences in the expression level of mTRPV3 in the plasma membrane.

In addition, we checked the heat-induced activation of TRPV3. The heat responses of HEK293T cells expressing both mTRPV3 and mTMEM79 were smaller than those expressing mTRPV3 alone (Fig. 1e–g and Supplementary Fig. 1a, b). TRPV3 has been implicated as a warmth sensor[29–31], and it becomes responsive to physiological temperatures after intensive stimulation with an initial high-temperature threshold close to 50 °C[32]. The temperature thresholds of TRPV3 channels were not affected by the co-expression of TMEM79 (Supplementary Fig. 1c, d), supporting the notion that TMEM79 exclusively modulates TRPV3 populations.

Given the similarities between TRPV4 and TRPV3 in warmth sensitivity and localization in the skin, we also examined whether co-expression of TMEM79 could affect TRPV4-mediated currents activated by GSK101. Interestingly, co-expression of TMEM79 did not alter 300 nM GSK101-induced currents (Supplementary Fig. 2a, b) or change the dose-dependence curves for GSK101 at both +60 and −60 mV (Supplementary Fig. 2c, d), suggesting that co-expression of TMEM79-specifically decreased currents evoked by TRPV3, but not by TRPV4.

### TMEM79 downregulates TRPV3 plasma membrane protein levels

We postulated above that TMEM79 reduced TRPV3 currents by regulating the functionality of TRPV3, perhaps by reducing the quantity of the protein. To address this, we performed Western blotting after labeling cell surface proteins with biotin. Given the fact that reliable anti-TRPV3 antibodies are hardly available, we used tagged proteins. By overexpressing an N-terminal myc-tagged mTRPV3 (myc-TRPV3), a C-terminal Flag-tagged mTMEM79 (TMEM79-flag), or transfecting both plasmids into HEK293T cells, we found that both the cell surface expression and total protein levels of TRPV3 were significantly decreased by co-expression of TMEM79 without affecting *Trpv3* mRNA levels (Fig. 2a, b and Supplementary Fig. 3a). It should be noted that the reduction of TRPV3 protein was more prominent in the plasma membrane (Fig. 2a, b). Moreover, we evaluated whether TMEM79 affected myc-tagged TRPV3-mediated currents by performing whole-cell patch–clamp experiments. Compared with myc-TRPV3 alone ($590.8 \pm 88$ pA/pF at +60 mV and $255.3 \pm 61.5$ pA/pF at −60 mV, $n = 20$), 1 mM 2-APB induced significantly smaller TRPV3 currents with co-expression of TMEM79 ($280 \pm 58.7$ pA/pF at +60 mV and $88.2 \pm 39.2$ pA/pF at −60 mV, $n = 14$; Supplementary Fig. 3b) while myc-tagged TRPV3-mediated currents were significantly smaller than untagged TRPV3-mediated currents. This result indicated that myc-mTRPV3 was functionally affected by the co-expression of TMEM79. To further confirm the level of plasma membrane TRPV3, we immunostained TRPV3 and cell surface membrane with antibodies for myc-tag and $Na^+/K^+$ ATPase, respectively, in HEK293T cells expressing myc-TRPV3 alone or expressing both myc-TRPV3 and TMEM79-flag. Similar to previous reports[33–35], we found significant co-localization of TRPV3 with $Na^+/K^+$ ATPase at the plasma membrane when TRPV3 was expressed alone (Fig. 2c, top panels). In contrast, co-expression of TMEM79 resulted in a broad distribution of TRPV3 in the cytoplasm in most cells (Fig. 2c, bottom panels), whereas the level of the TRPV3 signal remaining in the plasma membrane was lower. The co-transfected cells with robust cytoplasmic myc-TRPV3 staining contained vacuole-like structures (Fig. 2c, arrows). The above observation aligned with the results obtained from the whole-cell patch-clamp recordings and biotinylation assays, in which co-expression of TMEM79 reduced TRPV3-mediated currents by decreasing TRPV3 expression in the plasma membrane. These results demonstrated that co-expression of TMEM79 decreases TRPV3-mediated currents by reducing TRPV3 expression levels mainly in the plasma membrane.

### TMEM79 traps TRPV3 in the ER and causes TRPV3 degradation in the lysosomes

To further clarify the mechanisms governing the reduction of TRPV3 in the plasma membrane, we first evaluated the intracellular localization of TRPV3 combined with an antibody against calnexin, an ER organelle marker (Fig. 3a). We consistently observed high levels of TRPV3 protein in the plasma membrane with little co-localization with calnexin in cells expressing TRPV3 alone (Fig. 3a, top panels). In contrast, TRPV3 displayed predominant localization in the ER when TMEM79 was introduced (Fig. 3a, bottom panels), suggesting that TMEM79 caused accumulation of TRPV3 in the ER, consistent with TMEM79 being an ER-resident protein as shown in a previous report[36]. The results from Fig. 2a, b, and Supplementary Fig. 3a, showed that total TRPV3 protein levels were reduced by co-expression of TMEM79, which could be caused by either reduced protein synthesis or increased degradation of TRPV3. Therefore, we investigated those possibilities by assessing additional organelle localization and protein levels of TRPV3 with related inhibitors. We first checked the localization of TRPV3 with a lysosomal marker, lysosomal-associated membrane protein 1 (LAMP-1) (Fig. 3b). The colocalized signal was not apparent in the cells transfected with *myc-Trpv3* alone. However, in the cells expressing myc-TRPV3 and TMEM79-flag, the intense myc-TRPV3 labeling accumulated beside the nuclei with clustered LAMP-1-positive compartments, indicating that TMEM79 could facilitate TRPV3 degradation through a lysosomal pathway. To confirm the involvement of the lysosomal pathway, we performed Western blotting in HEK293T cells that

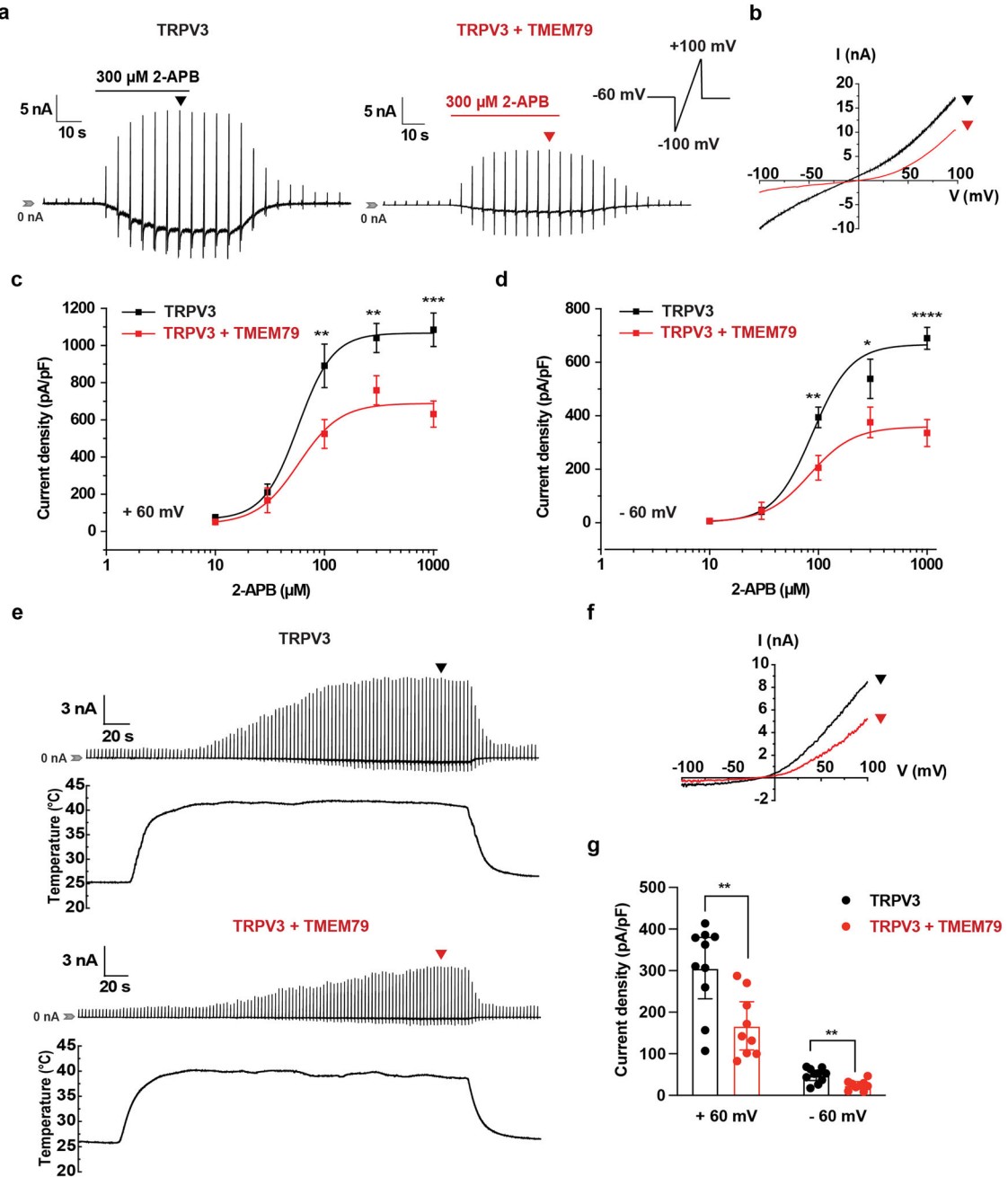

**Fig. 1 | TRPV3-mediated currents are reduced by co-expression of TMEM79.**
**a** Representative 2-APB-evoked (300 μM) current traces in a *Trpv3*-transfected HEK293T cell and a *Trpv3/Tmem79*-transfected HEK293T cell. Recordings were performed in a ramp-pulse protocol (−100 to +100 mV) every 3 s at a holding potential of −60 mV. Scale bars indicate current amplitudes (*y*-axis, nA) and time (*x*-axis, s). **b** Current–voltage (*I–V*) curves from the currents are shown in black and red in (**a**). **c**, **d** 2-APB dose-dependent curves in HEK293T cells transfected with *Trpv3* alone (black) or co-transfected with *Trpv3* and *Tmem79* (red) at +60 mV (**c**) and −60 mV (**d**). The EC$_{50}$ values were 57.4 ± 3.2 μM (black) and 57.8 ± 20.8 μM (red) at +60 mV, and 88.3 ± 13.6 μM (black) and 83.5 ± 18.9 μM (red) at −60 mV. Current densities (pA/pF) represent the largest values of the 2-APB-induced currents. The number of recordings in (**c**) and (**d**) is 4, 4, 7, 16, and 12 for cells expressing TRPV3

alone (black) and is 3, 5, 8, 23, and 15 for cells expressing TRPV3 and TMEM79 (red), respectively from low to high concentration of 2-APB. All curves were fitted with the Hill equation with a Hill coefficient (*n*) around 2. **e** Representative heat-evoked current traces in an *mTrpv3*-transfected HEK293T cell and a *Trpv3/Tmem79*-transfected HEK293T cell. **f** *I–V* curves from the currents are shown in black and red in (**e**). **g** Comparison of heat-induced current densities in HEK293 cells expressing mTRPV3 (306.3 ± 32.7 pA/pF at +60 mV and 48.2 ± 5.4 pA/pF at −60 mV, *n* = 10) or mTRPV3/mTMEM79 (167.1.3 ± 25.1 pA/pF at +60 mV and 24.4 ± 3.8 pA/pF at −60 mV, *n* = 9) at 60 ± mV. Statistics were performed by a one-tailed unpaired *t*-test. All error bars and data represent the mean ± SEM. *P < 0.05; **P < 0.01; ***P < 0.001, ****P < 0.0001.

overexpressed myc-TRPV3 with or without TMEM79-flag followed by treatment with two lysosome (chloroquine (CQ) and bafilomycin A1 (BafA1)) or proteasome (MG132 and Lactacystin) inhibitors for 8 h (Fig. 3c, d). We found that only CQ and BafA1 reversed the decrease in TRPV3 induced by TMEM79. This observation confirmed that TMEM79

promoted the degradation of TRPV3 through the lysosome pathway rather than through proteasome action (Fig. 3c, d). To examine the contribution of protein synthesis, we treated the cells with a protein synthesis inhibitor, cycloheximide (CHX, 100 μg/ml), 24 h after transfection. We then examined the amount of myc-tagged TRPV3 protein

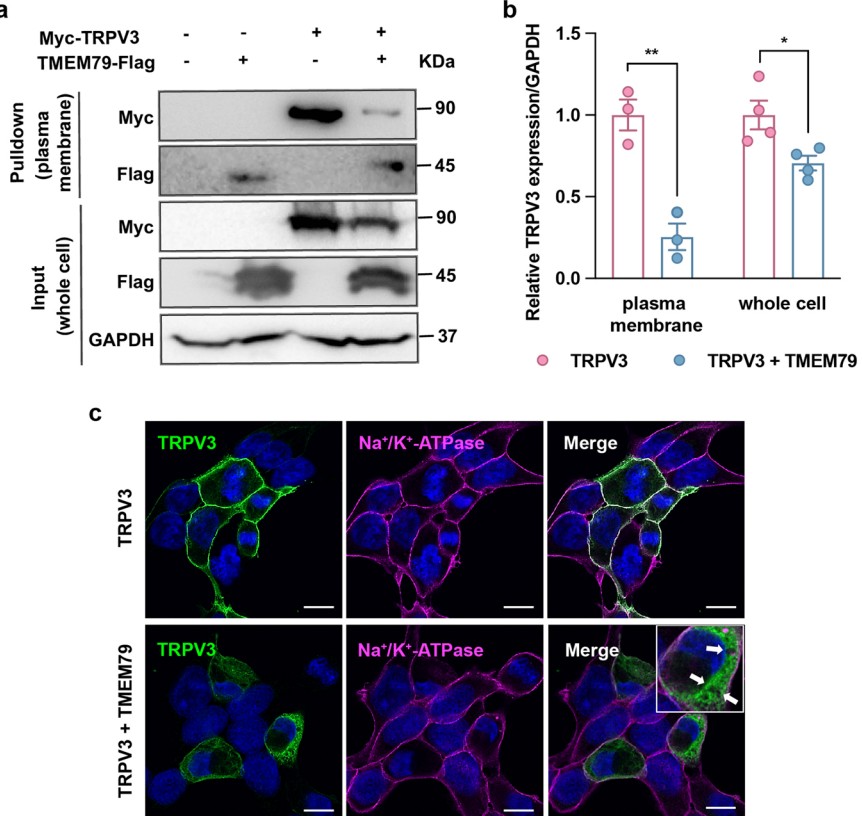

**Fig. 2 | TMEM79 decreases TRPV3 plasma membrane protein. a** HEK293T cells transiently transfected with empty vectors, *Tmem79-flag* alone, *myc-mTrpv3* alone, or both *myc-mTrpv3* and *Tmem79-flag* were biotinylated and immunoblotted with antibodies against the myc-tag and flag-tag at both the cell surface (pulldown) and whole-cell lysate (input) levels. **b** Quantitative analysis of the normalized band intensities of TRPV3 to GAPDH in HEK293T cells expressing myc-mTRPV3 alone (red circles) or co-expressing myc-mTRPV3 and mTMEM79-flag (blue circles) in the plasma membrane ($n = 3$) and whole-cell lysates ($n = 4$). Statistics were performed separately by a one-tailed unpaired *t*-test. All error bars represent the mean ± SEM. *$P < 0.05$; **$P < 0.01$. **c** Immunofluorescence images of TRPV3 localization ($n = 5$). HEK293T cells transiently transfected with *myc-Trpv3* alone, or both *myc-Trpv3* and *Tmem79-flag* were stained with antibodies against $Na^+/K^+$ ATPase (magenta) and myc-tag (green). Nuclei were stained with DAPI (blue). Myc-mTRPV3 alone clearly localizes with $Na^+/K^+$ ATPase (a cell membrane marker), while it displays cytoplasmic retention after the overexpression of mTMEM79. The arrows in the enlarged image indicate vacuole-like structures. Scale bars indicate 10 µm.

at 0, 3, 6, 12, and 24 h after starting the CHX treatment in cells with and without co-expression of TMEM79. We found that the normalized amount of TRPV3 was significantly smaller at 24 h CHX treatment in the cells co-expressing TMEM79 than in cells expressing TRPV3 alone (Fig. 3e, f). This result suggests that the reduction of TRPV3 is not caused by reduced protein synthesis, although it is hard to observe the effect of protein synthesis inhibition in overexpression systems like the one used in our HEK293T cells. Rather, the apparently greater reduction of TRPV3 in the experiment could be caused by the increased degradation occurring in the lysosomal pathway, as observed above. Together, the above results suggested that TMEM79 predominantly modulated the degradation of TRPV3 proteins in lysosomes.

## TRPV3 and TMEM79 form a complex

A physiological interaction between TRPV3 and TMEM79 implies a potential physical interaction between them. Therefore, we checked the possibility that they were colocalized in HEK293T cells (Fig. 4). HEK293T cells transfected with *myc-Trpv3* alone showed TRPV3 localization at the plasma membrane and no immunoreactivity for TMEM79-flag (Fig. 4a, top panels). In contrast, when *Tmem79-flag* was co-transfected, HEK293T cells showed overlapping signals between myc-TRPV3 and TMEM79-flag (Fig. 4a, bottom panels). We next tested whether TRPV3 and TMEM79 could be coimmunoprecipitated. Again, we transiently transfected HEK293T cells with empty pcDNA3.1⁺ vectors, *myc-Trpv3* alone, *Tmem79-flag* alone, or both plasmids together.

Anti-myc and anti-flag antibodies were utilized for the detection of TRPV3 and TMEM79 in transfected HEK293T cells. It was interesting that TRPV3 was specifically precipitated when co-transfected cell lysates were immunoprecipitated with an anti-flag antibody. TMEM79 was successfully identified when immunoblots of cell lysates were pulled down with an anti-myc antibody (Fig. 4b). We also carried out a proximity ligation (PLA) assay in collaboration with the plasma membrane marker ($Na^+/K^+$ ATPase) to further confirm the physical interaction between TRPV3 and TMEM79 (Fig. 4c, d). In the presence of both TRPV3 and TMEM79 in HEK293T cells, red punctate fluorescence was visible, indicating that the two proteins were in close proximity (within 40 nm). In addition, most PLA signals were clearly associated with the immunoreactivity of $Na^+/K^+$-ATPase, suggesting that the interaction was likely on or adjacent to the plasma membrane. Together, the above results demonstrated that TRPV3 did indeed interacts with TMEM79.

## Keratinocytes exhibit three distinct responses to 2-APB and GSK101

TRPV3 and TRPV4 are warm-temperature sensitive TRP ion channels that are highly expressed in skin keratinocytes with suggested key roles in skin physiology and pathophysiology[9]. TMEM79 is also detected in skin keratinocytes with contributions to skin barrier formation and itching sensations[24,26,27]. We, therefore, examined murine gene expression of *Trpv3*, *Trpv4*, and *Tmem79* by RT-PCR among skin tissues from the tail, ear, and back as well as the neck. The three genes

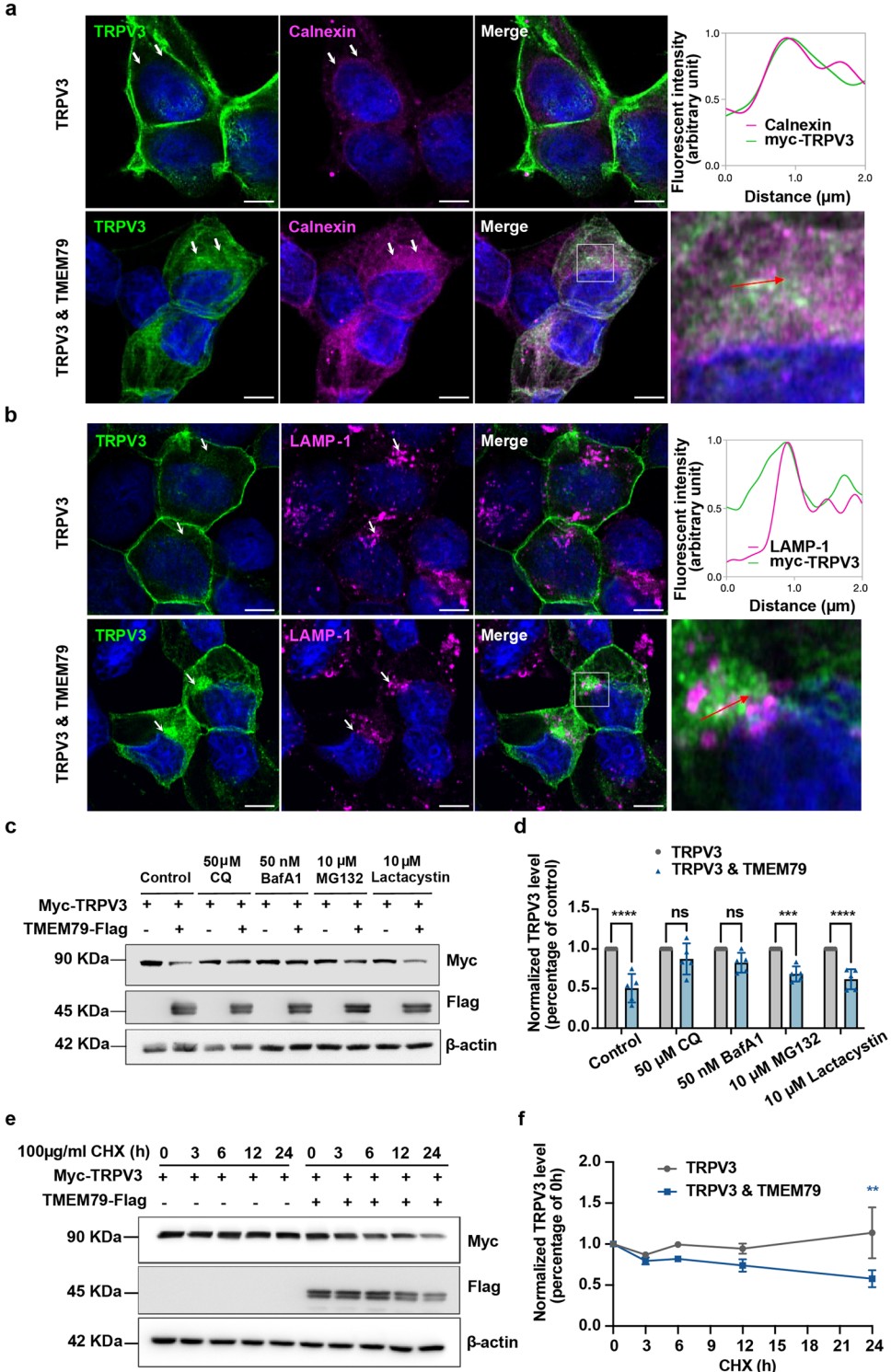

**Fig. 3 | Co-expression of TMEM79 alters the localization of TRPV3 and promotes TRPV3 degradation in the lysosomes. a, b** Representative super-resolution confocal images show the subcellular localization of myc-TRPV3 (green) and calnexin, ER marker (magenta, in **a**) or LAMP-1, lysosome marker (magenta, in **b**) in HEK293T cells. The cells expressing myc-TRPV3 alone show little co-localization of myc-TRPV3 and ER or lysosome marker staining, whereas cells co-expressing myc-TRPV3 and TMEM79-flag exhibit conspicuous cytoplasmic myc-TRPV3 signals co-localized with ER or lysosome marker. White arrows indicate the immunolocalization of calnexin or LAMP-1. The upper right graphs show the fluorescent intensity profile along the red arrowed lines in the white boxes of the enlarged images (bottom right). Nuclei were stained with DAPI (blue). All scale bars indicate 5 μm. $N = 3$ for (**a**) and (**b**). **c** HEK293T cells overexpressing myc-TRPV3 with or without

TMEM79-flag were treated with two lysosomes (CQ and BafA1) or proteasome inhibitors (MG132 and Lactacystin) for 8 h after 24 h of transfection. Protein expression was analyzed by Western blot experiments. **d** Quantification of normalized TRPV3 expression in (**c**), $n = 5$ for all. **e** HEK293T cells overexpressing myc-TRPV3 with or without TMEM79-flag were treated with 100 μg/mL CHX 24 h after transfection. Protein samples collected at 0, 3, 6, 12, and 24 h after CHX treatment were analyzed by western blot experiments. **f** Quantification of the intensity of TRPV3 corresponding bands in (**e**). The intensity of each band at 3, 6, 12, and 24 h after CHX treatment was normalized to β-actin and control bands at 0 h ($n = 4$). All statistics were performed by two-way ANOVA multiple comparisons. Error bars and data represent the mean ± SEM. **$P < 0.01$; ***$P < 0.001$, ****$P < 0.0001$.

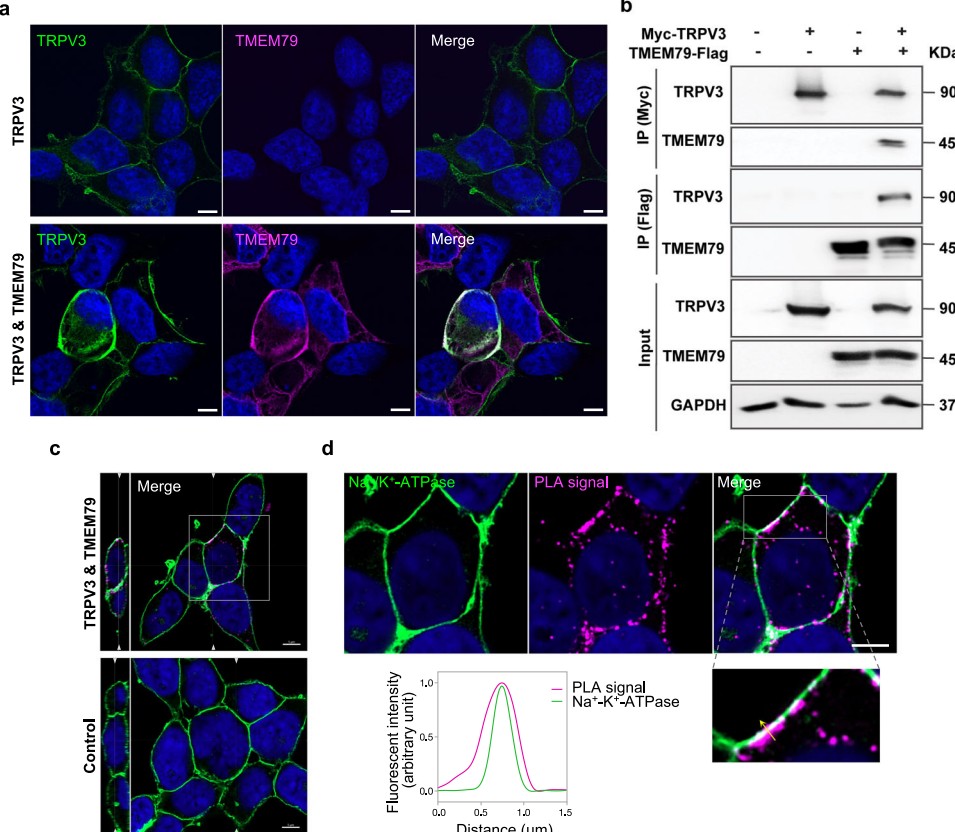

**Fig. 4 | TRPV3 and TMEM79 colocalize and bind to one another. a** Representative high-resolution confocal images of HEK293T cells expressing myc-mTRPV3 (green) alone or myc-mTRPV3 and mTMEM79-flag (magenta). Scale bars indicate 10 μm. **b** Coimmunoprecipitation (co-IP) of TRPV3 and TMEM79 in HEK293T cells. HEK293T cells transiently transfected with empty vectors, *myc-Trpv3* alone, *Tmem79-flag* alone, or both *myc-Trpv3* and *Tmem79-flag* were immunoprecipitated with anti-myc or anti-flag antibodies. Bands were detected using antibodies against the myc-tag and flag-tag by avoiding the IgG heavy chain around 50 kDa. Input immunoblots indicate TRPV3, TMEM79, as well as GAPDH bands in the whole-cell lysates before co-IP. **c** Representative super-resolution confocal images of the optical XY section of 0.16 μm with the PLA assay and immunostaining of Na⁺/K⁺- ATPase in HEK293T cells transfected with empty vectors (control) or *Trpv3/ Tmem79*. The magenta puncta indicate the sites of interaction between TRPV3 and TMEM79 in HEK293 cells. Orthogonal projections of YZ plane images are shown on the left side. The arrowheads indicate the position of the optical XY section (left) and YZ section (right). **d** Higher magnification images of the indicated box in (**c**) with cells expressing TRPV3 and TMEM79. Note that PLA signals are mainly located at or adjacent to the Na⁺/K⁺-ATPase immunoreactivity (plasma membrane). The graph shows the fluorescent intensity profile of the PLA signal and the Na⁺/K⁺- ATPase along the yellow arrowed line shown in the enlarged image. Nuclei were stained with DAPI (blue). Scale bars in **c** and **d** indicate 5 μm. All experiments were repeated three times.

were detected in these tissues (Supplementary Fig. 4a). Given the advantages of the hairless tail, we isolated primary keratinocytes from the tail epidermis (Fig. 5a and Supplementary Fig. 4a). The purity of keratinocytes was found to be nearly 100% with a keratin 14 marker (Fig. 5a). We afterward performed whole-cell patch-clamp recordings to assess the responses of primary skin keratinocytes to chemical stimuli. We applied 300 μM 2-APB, and subsequently 1 μM GSK101, to a single keratinocyte with voltage ramps from −100 to +100 mV at a holding potential of −60 mV. Following stimulation, we characterized the three types of keratinocytes from wild-type mice based on their different responses to 2-APB and GSK101 (Supplementary Fig. 4b). Both 2-APB and GSK101 induced substantial inward and outward currents. That is, 12.7% and 3.2% of keratinocytes showed responses to 2-APB alone or GSK 101 alone, respectively, while 84.1% of cells exhibited responses to both chemicals. This implies that most keratinocytes functionally express both TRPV3 and TRPV4.

### TMEM79 regulates TRPV3-mediated currents and calcium influx in primary skin keratinocytes

We concluded above that TMEM79 reduced 2-APB-induced TRPV3 currents in HEK293T cells. To clarify the effects of co-expression of TMEM79 in primary keratinocytes, we compared 300 μM 2-APB-evoked and 1 μM GSK 101-evoked currents in keratinocytes lacking

TMEM79 and wild-type littermate mice (the absence of *Tmem79* mRNA was confirmed, Fig. 5b). Consistent with the results in HEK293T cells, 300 μM 2-APB induced significantly larger currents in TMEM79⁻/⁻ keratinocytes compared with those from the wild-type, whereas GSK101-induced currents did not differ between them (Fig. 5c–e). The average current densities activated by 2-APB in wild-type and TMEM79⁻/⁻ keratinocytes were 86.8 ± 9.5 and 160.7 ± 18.6 pA/pF at +60 mV, and 75.5 ± 6.8 and 120.3 ± 12.4 pA/pF at −60 mV, respectively (Fig. 5d). The average current densities activated by GSK101 in wild-type and TMEM79⁻/⁻ keratinocytes were 122.4 ± 10.5 and 123.7 ± 9.6 pA/ pF at +60 mV, and 64.1 ± 6.0 and 56.7 ± 5.4 pA/pF at −60 mV, respectively (Fig. 5e). Considering that 2-APB is not a specific agonist for TRPV3, we also examined channel activity in TRPV3⁻/⁻ keratinocytes (Fig. 5c, bottom). As expected, 2-APB did not elicit obvious currents, whereas GSK101 induced comparable currents (166.8 ± 36.2 pA/pF at +60 mV and 82.1 ± 19.7 pA/pF at −60 mV) compared with wild-type and TMEM79⁻/⁻ keratinocytes, indicating that 2-APB-induced currents are derived from TRPV3 (Fig. 5d, e).

For further evaluation, calcium imaging experiments were conducted to study TRPV3-mediated activities by measuring changes in intracellular calcium concentrations ([Ca²⁺]ᵢ) in Fura-2-loaded keratinocytes stimulated with a TRPV3 agonist cocktail (300 μM 2-APB and 300 μM carvacrol) or heat (42 °C at peak) (Fig. 5f, g and

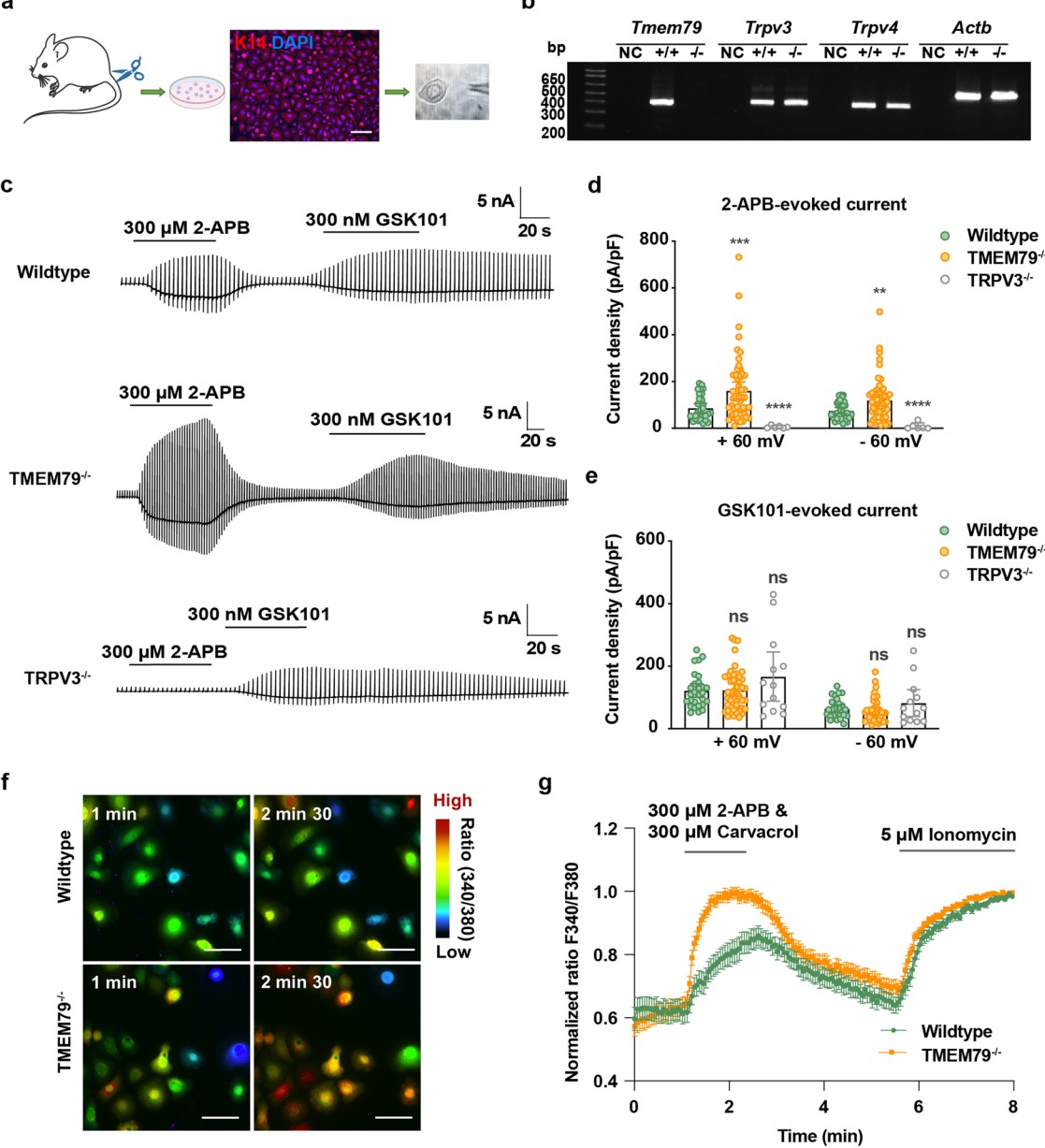

**Fig. 5 | Increased TRPV3-mediated currents and intracellular Ca²⁺ concentrations in skin keratinocytes lacking TMEM79. a** Preparation of primary keratinocytes from tails of male mice (3–5 weeks old). The purified keratinocytes were stained with a keratinocyte marker, keratin 14 (K14, red), and DAPI (blue). The scale bar indicates 10 μm. The Figure was partly modified using Servier Medical Art under a Creative Commons Attribution 3.0 license. **b** RT-PCR for *Tmem79*, *Trpv3*, *Trpv4*, and *Actb*. Bands were amplified from tail primary keratinocytes (KCs) of wild-type and TMEM79⁻/⁻ mice. All mice samples were genotyped before experiments. **c** Representative current traces evoked by 2-APB (300 μM) and continuous GSK101 (300 nM) in keratinocytes from wild-type (top), TMEM79⁻/⁻ (middle), and TRPV3⁻/⁻ (bottom) mice. Recordings were performed in a ramp-pulse protocol (−100 to +100 mV) every three seconds at a holding potential of −60 mV. Scale bars indicate current amplitudes (*y*-axis, nA) and time (*x*-axis, s). **d** Comparison of 300 μM 2-APB-induced current densities in primary keratinocytes from wild-type (green circles,

$n = 31$), TMEM79⁻/⁻ (orange circles, $n = 55$), and TRPV3⁻/⁻ (gray circles, $n = 6$) mice. **e** Comparison of 300 nM GSK101-induced current densities in primary keratinocytes from wild-type (green circles, $n = 28$), TMEM79⁻/⁻ (orange circles, $n = 47$), and TRPV3⁻/⁻ (gray circles, $n = 13$) mice. Statistics were performed by a two-tailed unpaired *t*-test. All data represent the mean ± SEM, and all error bars indicate a 95% confidence interval for the mean. **$P < 0.01$; ***$P < 0.001$; ****$P < 0.0001$; ns $P > 0.05$. **f** Representative fluorescence images showing changes in the Fura-2 ratios in primary keratinocytes from wild-type and TMEM79⁻/⁻ mice in response to 300 μM 2-APB/carvacrol. Scale bars indicate 50 μm. **g** Fura-2 ratio (340/380) induced by a TRPV3 agonist cocktail (300 μM 2-APB with 300 μM carvacrol) in primary keratinocytes derived from wild-type (green) and TMEM79⁻/⁻ (orange) mice. Fura-2 ratios were normalized to 5 μM ionomycin; $n = 96$ (green) and 181 (orange) cells, respectively, from three times repeat. Error bars indicate a 95% confidence interval for the mean.

---

Supplementary Fig. 5), an approach that had been commonly used in previous reports[14,37]. Five μM ionomycin was introduced after each stimulus to normalize [Ca²⁺]ᵢ values. Strikingly, [Ca²⁺]ᵢ in individual cells was greater in keratinocytes from TMEM79⁻/⁻ mice compared with wild-type mice upon chemical and temperature stimulation, confirming the increased activity of TRPV3 channels in TMEM79⁻/⁻

keratinocytes. Together, the above data corroborated that TMEM79 is required for regulating TRPV3 activity in skin keratinocytes.

**TMEM79 is involved in temperature sensing**
TRPV3 reportedly plays a role in temperature sensation within an innocuous range. However, some studies have suggested that TRPV3

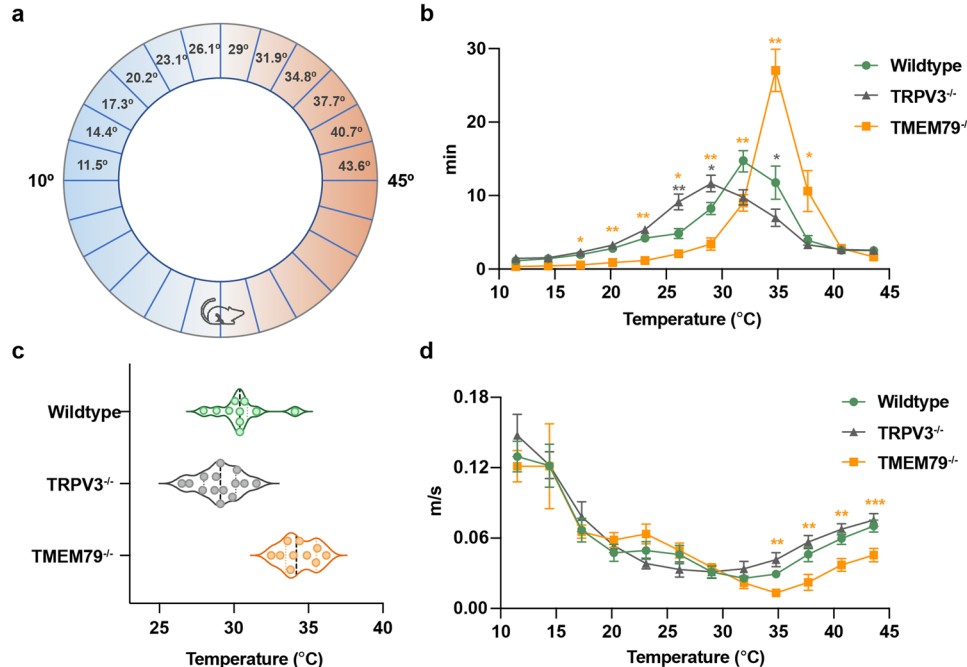

**Fig. 6 | TMEM79-deficient mice exhibit a strong preference for a warmer temperature compared with wild-type mice.** A thermal gradient ring was divided into two identical 12-zone halves ranging from 11.5 °C to 43.6 °C with 2.9 °C increments. Wild-type (green), TRPV3$^{-/-}$ (gray), and TMEM79$^{-/-}$ (orange) mice (male, 7-10-weeks old) were released at the middle position of the thermal gradient and were then allowed to freely move for 1 h. **b** "Spending time" of wild-type, TRPV3$^{-/-}$, or TMEM79$^{-/-}$ mice in each temperature zone during a 1 h recording. **c** Violin plots of preferred temperatures for wild-type (30.4 °C ± 0.5 °C), TRPV3$^{-/-}$ (29.1 °C ± 0.4 °C), or TMEM79$^{-/-}$ (34.4 °C ± 0.4 °C) mice. Mean values (gray dotted line) were calculated based on spent time. (d) "Moving speed" of wild-type, TRPV3$^{-/-}$, or TMEM79$^{-/-}$ mice within each temperature zone. In **b**–**d**, the $n = 10$, 14, and 10 for wild-type (green), TRPV3$^{-/-}$ (gray), and TMEM79$^{-/-}$ (orange) mice, respectively. Statistics in **b** and **d** were performed by a mixed-effects (two-way) ANOVA with Geisser–Greenhouse correction. Multiple comparisons were performed without correction. All data and error bars represent the mean ± SEM. $*P < 0.05$; $**P < 0.01$; $***P < 0.001$.

makes only a slight contribution to warmth perception and that the effects varied among different genetic backgrounds or genders[38]. We, therefore, wondered how the interaction between TRPV3 and TMEM79 could be involved in thermosensation. In our study, we used a thermal gradient ring assay by setting the temperature from 10 to 45 °C to avoid the corner effect (Fig. 6a). The apparatus was divided into two identical 12-increment zones, with each zone representing an interval of approximately 2.9 °C. "Spending time" or "moving speed" in each zone was recorded for one h. Different from wild-type mice that consistently showed a moderate preference for 29–34.8 °C (Fig. 6b, green), TRPV3$^{-/-}$ mice on a C57BL6 background exhibited a wider distribution of 26.1–34.8 °C (Fig. 6b, gray), which was consistent with previous studies and a report from our lab[7,8]. It was also interesting that TRPV3$^{-/-}$ mice and wild-type mice showed similar preferences at temperature zones lower than 23.1 °C and higher than 37.7 °C.

By analyzing the exact peak temperature, we observed that wild-type mice favored 30.4 ± 0.5 °C ($n = 10$) while the favored temperatures were shifted to 29.1 ± 0.4 °C in TRPV3$^{-/-}$ mice ($n = 14$), demonstrating that TRPV3 is involved in warm temperature perception (Fig. 6c). Interestingly, when we examined the thermal preferences of TMEM79$^{-/-}$ mice, they displayed a more restricted distribution than wild-type and TRPV3$^{-/-}$ mice (Fig. 6b, orange). TMEM79$^{-/-}$ mice spent nearly half their time in a temperature zone around 35 °C with a peak at 34.4 ± 0.4 °C ($n = 10$; Fig. 6b, c). We also examined the movements of mice every 20 min. From the data shown in Supplementary Fig. 6a–c, it was apparent that wild-type mice started discriminating the optimal temperature after 20 minutes (Supplementary Fig. 6a), while TRPV3$^{-/-}$ mice showed indifference to warm temperature within 40 min (Supplementary Fig. 6b). TMEM79$^{-/-}$ mice chose the preferred temperature within the first 20 min (Supplementary Fig. 6c). This interpretation was supported by Supplementary Fig. 6d, which shows that TMEM79$^{-/-}$

mice quickly chose to stay at their preferred temperature within 5 min. Moreover, we evaluated the moving speed in each temperature zone since mice were expected to move slower in regions where they felt more comfortable. Consistent with their preferred temperatures, wild-type mice exhibited the slowest speed in the 31.9–34.8 °C zones, whereas TRPV3$^{-/-}$ and TMEM79$^{-/-}$ mice moved slowest in the 23.1–31.9 °C zones and the 34.8 °C zone, respectively (Fig. 6d). Based on the above results, we speculated that the altered thermal selection of TMEM79$^{-/-}$ mice might be due to increased TRPV3 expression. This interpretation was supported by the fact that there was no difference in the time spent in a zone without a thermal gradient (Supplementary Fig. 6e).

## Discussion

The TRP superfamily of ion channels plays a variety of roles in physiology and pathophysiology. Therefore, it is important to understand their regulation because alterations might lead to the progression of diseases or physiological abnormalities. TRPV3 is a molecular sensor of warmth and is expressed abundantly in skin keratinocytes. Recently, its structure mechanism by heat stimulation was comprehensively introduced showing heat-induced conformational changes in numerous pore domains[39,40]. However, studies of TRPV3 modulation are limited. Here, we demonstrate that a membrane protein, TMEM79, is a negative regulator of TRPV3.

In HEK293T cells, co-expression of TMEM79 decreased TRPV3-mediated current amplitudes. The altered ion channel activity could involve either fewer numbers of TRPV3 ion channels in the plasma membrane or changes of the channel properties themselves. Based on the unchanged EC$_{50}$ values for the 2-APB effects in TRPV3- and TRPV3/TMEM79-expressing cells (Fig. 1c, d), we initially believed that TMEM79 decreased TRPV3 activity by reducing TRPV3 channel protein density

in the plasma membrane. This was supported by a biotinylation assay and co-immunostaining for TRPV3 with a plasma membrane marker (Fig. 2a). Those experiments demonstrated that the cell surface levels of TRPV3 protein were apparently lower when TMEM79 was introduced. Furthermore, we revealed that the reduction of TRPV3 protein levels caused by TMEM79 was associated with the lysosomal proteolysis pathway rather than proteasomal degradation or reduction of TRPV3 protein synthesis (Fig. 3c–f) by using several inhibitors that were extensively applied in many studies[41–43]. Consistent with the above roles for TMEM79 in modulating TRPV3, we also substantiated that TMEM79 physically interacted with TRPV3. However, it is still not clear which regions of either protein contribute to that interaction. Further studies will be required to assess the binding sites by making chimeras and narrowing down the protein motifs.

Given the high expression of TRPV3, TRPV4, and TMEM79 in skin keratinocytes, we opted to isolate primary keratinocytes from mouse skin and perform whole-cell patch–clamp experiments similar to that done in HEK293T cells. The functional expression of both TRPV3 and TRPV4 has been identified in previous studies[9,13,21,44,45]. In this work, we newly characterized three types of keratinocytes based on their response to either 2-APB, GSK101, or both reagents, suggesting distinct expression patterns of TRPV3 and TRPV4 in those cells (Supplementary Fig. 4b). Additionally, over 90% of keratinocytes responded to 2-APB in our study, which is consistent with the high expression of TRPV3 in keratinocytes, but contradictory to several findings that only small populations of keratinocytes or HaCaT cells responded to heat or 2-APB[9,35]. The difference between the previous data and the data found here could be due in part to the fact that we performed whole-cell patch-clamp experiments in a calcium-free bath and pipette solutions in HEK293T cells and primary keratinocytes, whereas calcium-containing solutions were applied in most other studies. TRPV3 is unique in its sensitization to repetitive heat or chemical stimulation through both calcium-dependent and -independent mechanisms. Several studies have reported that depletion of calcium in intracellular and/or extracellular solution potentiates TRPV3 currents[13,30,46,47]. These aspects might explain why we observed more robust currents in keratinocytes. Although we used $Ca^{2+}$-free extracellular solution to observe constant TRPV3-mediated currents, $Ca^{2+}$ could be involved in the interaction between TRPV3 and TMEM79 under physiological conditions.

One previous study suggested that TRPV3 activation in keratinocytes is important in itch transmission[48]. Indeed, TRPV3 protein levels were increased in skin biopsies of several chronic itch diseases such as atopic dermatitis, rosacea, and psoriasis[37,49,50]. In particular, intradermal injection of carvacrol, a TRPV3 agonist, induced robust scratching behavior[20]. In contrast, pharmacological application of TRPV3 antagonists helped to alleviate scratching behavior in several itching models[20,21,51]. Similarly, the lack of TMEM79 in keratinocytes increased itching behaviors in mice[24]. Therefore, our study may explain the mechanism of itching based on the interaction between TRPV3 and TMEM79.

In mammals, it was believed that thermal information was conveyed from the skin through sensory nerve endings located in the epidermis to the peripheral and CNS and that thermosensitive TRP channels are the key mediators[1,4,52]. In addition, keratinocytes were suggested to contribute to thermosensation likely via warmth sensors such as TRPV3[5]. Our early study showed that TRPV3 and ATP were coupled with the activation of purinergic P2X receptors to regulate cell-cell communication among keratinocytes[10]. Activation of TRPV3 in keratinocytes also induced secretion of diffusible messengers such as nitric oxide (NO), prostaglandin E2 ($PGE_2$) and TGF-α, which could help transmit temperature information[14,17,53]. However, the TRPV3 knockout mice did not always show impaired temperature sensation in vivo, particularly in the innocuous range. In the linear gradient assay, TRPV3-deficient mice stayed mostly at cooler temperature without affecting

their noxious heat and cold preference although more recent studies argued that temperature preference in TRPV3-deficient mice varied between different genetic backgrounds[7,38,54]. In our study, we attested to a role for TRPV3 in warm temperature sensation with a Thermal Gradient Ring that holds some advantages by eliminating the mice's stress and corner effects. Moreover, it was surprising to find the impact of TMEM79 deletion on mouse thermosensation in which TMEM79-deficient mice exhibited a strong preference for warmer temperature. Based on the elevated TRPV3 activities in keratinocytes lacking TMEM79, we speculate that the lack of TMEM79 from keratinocytes increases membrane TRPV3 expression, which in turn further induces larger currents and transmits temperature information directly or indirectly to the CNS.

TMEM79 was proposed to be a putative glutathione transferase (MAPEG) that detoxified reactive oxygen species[24]. But to date, there has been no direct evidence to support the concept that TMEM79 could catalyze the conjugation of glutathione or other related catalytic reactions. Although the exact biochemical function of TMEM79 remains unclear, our results suggest that TMEM79 is involved in post-translational modification processes such that it promotes TRPV3 degradation. This interpretation is supported by a prediction in the human reference interactome (HuRI) map (http://www.interactome-atlas.org/) in which TMEM79 potentially interacts with a series of proteins, including vesicle-trafficking protein (SEC22a), testis-expressed protein 264 (TEX264, which is important for ER remodeling), translocating chain-associated membrane protein 1-like 1 (TRAM1L1, which is required for translocation of proteins from the ER), stress-associated endoplasmic reticulum protein 1 (SERP1), and E3 ubiquitin-protein ligase (RNF8)[55].

Overall, our study reveals unique interactions between TRPV3 and TMEM79 and provides further evidence that TRPV3 is required for temperature sensing.

## Methods

### Animals and genotyping

Male heterozygous TMEM79-deficient (TMEM79[+/−]) mice on a C57BL/6NCr background were kindly provided by Dr. Takeshi Matsui and Dr. Masayuki Amagai (Laboratory for Skin Homeostasis, RIKEN, Japan). Homozygous TMEM79-deficient (TMEM79[−/−]) mice were generated by mating male TMEM79[+/−] and female wild-type mice for two generations. Wild-type, TMEM79[+/−], and TMEM79[−/−] mice were genotyped with primers *Tmem79-Intron1-F*, *Lar3-Universal-R*, and *Tmem79-Exon2-R* (Supplementary Table 1). The recombinant Taq™ DNA Polymerase (Takara, R001) kit was used in a 10 µL PCR reaction with a thermal profile starting at 94 °C for 2 min, followed by 35 cycles at 94 °C for 15 s, 55 °C for 30 s, 72 °C for 30 s, and a final one min extension at 72 °C. All mice were maintained under SPF conditions in a controlled environment (12-h light/dark cycle with free access to food and water, 25 °C, and 50–60% humidity). All procedures were approved by the Institutional Animal Care and Use Committee of the National Institute of Natural Sciences and carried out according to the National Institutes of Health and National Institute for Physiological Sciences guidelines.

### Cell culture

HEK293T cells were maintained at 37 °C and 5% $CO_2$ in a high glucose D-MEM culture medium (Wako, 044-29765) supplemented with 10% fetal bovine serum (Gibco, 10437-028), 2 mM GlutaMAX (Gibco, 35050-061), 50 units/mL penicillin, and 50 µg/mL streptomycin (Gibco, 15140-122). Cell passage was performed every 2 or 3 days.

### DNA construction

Plasmid 3xFLAG-m*Tmem79* was generously provided by Dr. Takeshi Matsui and Dr. Masayuki Amagai. To obtain a plasmid without tags for patch-clamp experiments, mouse *Tmem79* cDNA (NM_024246) was subcloned into the pcDNA3.1(+) vector (Invitrogen, V79020). In

detail, 3Xflag tag repeats at N-terminus were deleted by primers (Supplementary Table 1) *Tmem79-KpnI-F* and *Tmem79-NotI-R* followed by adding restriction sites KpnI and NotI at two ends. The phusion high-fidelity DNA polymerase (NEB, M0530S) was used with a PCR thermal profile starting at 98 °C for 30 s, followed by 10 cycles at 98 °C for 10 s, 60 °C for 30 s, 72 °C for 30 s, another 25 cycles at 98 °C for 10 s, 71 °C for 30 s, 72 °C for 30 s and a final 10 min extension at 72 °C. This gene was further inserted into pcDNA3.1 (+) backbone. Mouse *Trpv3* cDNA (AF510316) was cloned into the same pcDNA3.1(+) vector with or without a c-myc tag for electrophysiological or molecular experiments.

### Primary mouse keratinocyte isolation and culture

Mice at 4–5 weeks of age were first anesthetized with isoflurane then euthanized by cervical dislocation. Tails were then isolated and the skin was peeled off. The skin from each tail was dissected into four pieces and incubated overnight (O/N) on a cold room rotator in 4 mg/mL DISPASE II (Wako, 383-02281) in customized MCDB153 medium (CSR, CK015) containing 5 µg/mL insulin (Sigma, I1882), 0.4 µg/mL hydrocortisone (Sigma, H0888), 10 µg/mL transferrin (SCIPAC, P158-5), 14.1 µg/mL phosphorylethanolamine (Sigma, P0503), 10 ng/mL epidermal growth factor (Sigma, E4127), 25 µg/mL gentamicin (Gibco, 15710064), 50 units/mL penicillin, 50 µ/mL streptomycin, and 40 µg/mL bovine pituitary extracts (Kyokuto, 20200). After 12–16 h of incubation, the epidermis was detached from the dermis, and placed in 0.25% trypsin (Gibco, 15050065) for 20 min at room temperature (RT) with the basal layer down. Keratinocytes were next mechanically dissociated with dissecting forceps and filtered through a 100 µm cell strainer. The collected cells were incubated at 37 °C in 5% $CO_2$ for further patch–clamp and molecular studies. Primary keratinocytes were maintained in the above culture medium for a maximum of one week with fresh medium replaced daily.

### Transient transfection

HEK293 cells ($5 \times 10^5$) were seeded onto a 3.5 cm-cell culture dish one day before transfection. After O/N incubation at 37 °C in 5% $CO_2$, cells were transfected with Lipofectamine reagent (Invitrogen, 18324020), PLUS reagent (Invitrogen, 11514015), Opti-MEM I (Gibco, 31985070), and a total of 1 µg DNA at a 1:1 ratio from the paired combination of pcDNA3.1+ vector, (myc)-mTrpv3, or mTmem79-(3xflag) plasmids. For the patch-clamp assay and immunostaining, cells were redistributed on 12 mm micro-cover glasses (Matsunami) after three h of incubation.

### Whole-cell voltage-clamp

For HEK293T cells, patch-clamp recordings were performed within 24–30 h of the transfection. For primary skin keratinocytes, patch-clamp recordings were performed 2–4 days after isolation. A single separated cell was selected for each recording. Patch electrodes (King Precision Glass, 8250) were fabricated using a micropipette puller (Sutter, P-97) with a resistance of 3-5 MΩ. A standard extracellular 2 mM $Ca^{2+}$ bath solution (140 mM NaCl, 5 mM KCl, 2 mM $CaCl_2$, 2 mM $MgCl_2$, 10 mM glucose and 10 mM HEPES, pH 7.4 adjusted with NaOH) or a $Ca^{2+}$-free bath solution (140 mM NaCl, 5 mM KCl, 2 mM $MgCl_2$, 10 mM glucose, 5 mM EGTA and 10 mM HEPES, pH 7.4 adjusted with NaOH) was applied for studies of physiological interactions among TRPV4/TMEM79 and TRPV3/TMEM79 in HEK293 cells. The $Ca^{2+}$-free bath solution was also used in keratinocyte recordings. A standard intracellular pipette solution (140 mM KCl, 5 mM EGTA, and 10 mM HEPES, pH 7.4 adjusted with KOH) was utilized for all recordings. After establishing a whole-cell configuration, the current was recorded from −100 mV to +100 mV voltage-ramps every 3 s with a −60 mV holding potential, by applying 2-APB (Sigma) or GSK101 (Sigma) to the bath solution or heat. Series resistance and membrane capacitance were corrected. The data were recorded at 10 kHz (pCLAMP)

and filtered at 5 kHz (Clampfit) through which current density (pA/pF) was calculated.

### Ca²⁺ imaging

Primary mouse skin keratinocytes from wild-type or TMEM79−/− mice were seeded onto coverslips and incubated at 37 °C with fresh medium that was replaced daily. All recordings were performed 2–4 days after isolation. Five micrometer fura-2-acetoxymethyl ester (Fura2-AM, Invitrogen, F1201) was added one h prior to recordings. The 300 µM 2-APB and carvacrol-containing or pre-heated bath solutions (140 mM NaCl, 5 mM KCl, 2 mM $CaCl_2$, 2 mM $MgCl_2$, 10 mM glucose and 10 mM HEPES, pH 7.4 adjusted with NaOH) were prepared before recording. The coverslip was first mounted in a chamber connected to a gravity flow system, allowing for delivery of different solutions, followed by perfusion of bath solution for about one minute. Subsequently, a 300 µM 2-APB/carvacrol cocktail or a heated bath solution was perfused, followed by 5 µM ionomycin (Sigma, I0634). Fura-2 was excited at 340 nm and fluorescence was captured at 380 nm every three seconds. Data were obtained and analyzed using NIS-Elements AR (Nikon) and Microsoft Excel. To compare the $Ca^{2+}$ imaging traces from wild-type and TMEM79−/− keratinocytes, all data were displayed as a normalized ratio (F340/F380) relative to ionomycin.

### Co-immunoprecipitation

Twenty-four h after transfection, HEK293 cells were gently rinsed with cold 1× PBS and collected in 600 uL RIPA buffer (Thermo Scientific, 89900) supplemented with a protease inhibitor and a phosphatase inhibitor. Next, the cell lysate was placed on ice for 30 min with pipetting every 10 min followed by centrifugation at 12,000×*g* for 30 min. The following steps were all performed at 4 °C. Each supernatant sample was first cleared for 1 h with 1 µg of off-target monoclonal IgG1 antibody produced in mice then transferred into 50 µL of a pre-washed, Protein G Mag Sepharose bead (Cytiva, 28951379) slurry for 30 min of pre-cleaning. The cell lysates were next incubated with 2 µg of mouse anti-MYC (MBL, M047-3) or mouse anti-FLAG (Sigma, F3165) O/N, and the bead pellet was discarded. On the second day, the antibody-antigen complex was precipitated with 25 µL of fresh protein G beads for 3 h with subsequent washing in lysis buffer three times. The complex was then eluted from the beads in 40 µL of 1× sample buffer (Bio-Rad) by heating at 50 °C for 10 min. Five microlitres of each sample was loaded for Western blotting. Mouse anti-MYC and rabbit anti-FLAG (Santa Cruz Biotechnology, sc-807) antibodies were used to probe for TRPV3 and TMEM79 bands, respectively.

### Biotinylation of cell surface proteins

A biotinylation assay was performed in HEK293 cells 24 h after transfection as co-IP. Cells were first washed once with PBS then incubated twice with 0.5 mg/mL EZ-Link™ Sulfo-NHS-Biotin (Thermo, 21217) for 10 min at 37 °C. Next, the biotinylated cells were rinsed once with cold quenching buffer (100 mM glycine in PBS at pH 7.3) and then washed with PBS. The cells were collected in 200 µL RIPA buffer supplemented with protease inhibitor and lysed on ice for 30 min with pipetting, from which 150 µL of lysate was precipitated using 10 µL of Dynabeads MyOne Streptavidin T1 (Invitrogen, 65601) O/N at 4 °C while the remaining 50 µL of cell lysate was used as an input sample. On the following day, the beads were denatured in 30 µL of 1× sample buffer for 5 min at 95 °C after cleaning. Four microlitres of pull-down samples and 10 µL of input samples were electrophoresed on an 8% SDS-polyacrylamide gel. Cells transfected with empty vectors were used as a control.

### Western blot

Protein samples from both HEK293 cells and primary keratinocytes were lysed in RIPA buffer with protease/phosphatase inhibitor and

denatured at 95 °C for 5 min or 50 °C for 10 min. Equal amounts of protein were loaded and electrophoresed on 8% SDS-polyacrylamide gels for 1.5 h at 120 V. The proteins were next transferred to a PVDF membrane in a cold room for 2 h at 120 V. After blocking for 1 h with BLOCK ACE (KAC, UKB80) at RT, the membrane was incubated with antibodies to detect target bands. The primary antibodies included mouse anti-FLAG (Sigma, F3165, 1:2000), rabbit anti-FLAG (Santa Cruz, sc-807, 1:1000), mouse anti-MYC (MBL, 1:1000), and HRP-conjugated anti-GAPDH (Cell Signaling, 3683, 1:1000). The secondary antibodies consisted of anti-mouse IgG (Cell Signaling, 7076, 1:10000) and anti-rabbit IgG (Cell Signaling, 7074, 1:10,000). All proteins were incubated with an ECL kit (Cytiva), and the blots were imaged using a LAS-3000 mini (Fujifilm). Quantification was conducted with ImageJ.

### Immunofluorescence

Transfected HEK293 cells grown on coverslips were fixed with chilled methanol for 10 min on ice followed by three PBS washes and subsequently blocked with 10% goat serum (Sigma) for one h at RT. The fixed cells were incubated O/N at 4 °C with the following primary antibodies: rabbit anti-sodium potassium ATPase (Abcam, ab76020, 1:500), mouse anti-MYC (MBL, 1:200), rabbit anti-FLAG (Santa Cruz, 1:50), mouse anti-LAMP1 (Santa Cruz, sc-20011, 1:100), or mouse anti-calnexin (Santa Cruz, sc-46669, 1:100). On the second day, cells were washed with PBS three times for 5 min each and incubated with secondary antibodies at room temperature for 1.5 h. The following secondary antibodies were used: goat anti-mouse IgG (A-11029, Alexa 488 conjugated, Invitrogen), goat anti-mouse IgG (A-11032, Alexa 594 conjugated, Invitrogen), goat anti-rabbit IgG (A-11034, Alexa 488 conjugated, Invitrogen), goat anti-rabbit IgG (A-11037 or Alexa 594 conjugated, Invitrogen). All secondary antibodies were diluted 1:1000. The coverslips were finally mounted after incubation with DAPI for 10 min at RT. Subcellular immunocytochemical localization was observed at super-resolution level by Zeiss Axio Observer 7 and LSM 880 confocal unit with Airyscan module or Zeiss Axio Observer Z1 and LSM 800 confocal unit with Airyscan module (Carl Zeiss), and Plan Apochromat 63×/1.40 NA Oil DIC M27 objective (Carl Zeiss)[56]. To obtain super-resolution images (~140 nm in the XY plane and 200–350 nm in the XZ or YZ planes), cells were observed with optical slices of 160 nm and images were processed with Airyscan processing using ZEN Blue edition 3.5 software (Carl Zeiss).

### Duolink PLA assay

HEK293 cells were transfected with mTRPV3 alone or a combination of mTRPV3 and mTMEM79 as described above. After 24 h, cells were fixed in 4% paraformaldehyde (PFA; Wako) for 10 min, permeabilized by PBST (0.25% triton in PBS) for 15 min at RT and blocked with Duolink Blocking Solution (Sigma) for one h at 37 °C. Cells were next incubated with primary mouse anti-TRPV3 (Abcam, ab85022, 1:200) and rabbit anti-TMEM79 (Novus, NBP1-59832, 1:200) antibodies O/N at 4 °C. On the second day, the cells were washed twice with buffer A (0.01 M Tris, 0.15 M NaCl, and 0.05% Tween 20, pH 7.4 adjusted with HCl), and incubated with Duolink PLA Probes anti-mouse PLUS (Sigma, DUO92001) and anti-rabbit MINUS (Sigma, DUO92005) for one h at 37 °C. Thereafter, ligation and amplification were performed with the red Duolink in situ detection reagents kit (Sigma, DUO92008). The plasma membrane was stained with rabbit anti-Na⁺/K⁺ ATPase. Cells were then washed twice with buffer B (0.2 M Tris and 0.1 M NaCl, pH 7.5 adjusted with HCl) for 10 min each, rinsed with 0.01% buffer B for 1 min, and mounted with Duolink in situ mounting medium with DAPI (Sigma, DUO82040). Images were obtained using LSM 800 Airyscan (Carl Zeiss).

### Cycloheximide chase assay and inhibitor treatment

HEK293T cells in 3.5-cm cell culture dishes were transfected with *myc-TRPV3* with or without *TMEM79-flag* plasmids and re-seeded into

a 24-well plate. The cells were divided equally between five wells for each dish 3 h later. After transfection for 24 h, the supernatant medium was replaced with DMSO (control), 100 μg/mL CHX (Wako, 037-20991), 50 μM CQ (Sigma, C6628), 50 nM BafA1(Sigma, B1793), 10 μM MG132 (Wako, 139-18451), or 10 μM Lactacystin (BLK, 0460)-containing fresh medium. For CHX, the cells were lysed at the indicated time points and analyzed by Western blotting. For lysosome and proteasome inhibitors, cells were lysed after 8 h of treatment before analysis.

### RT-PCR and RT-quantitative PCR

Total RNA was extracted from primary keratinocytes with Sepasol-RNA I Super G (Nacalai Tesque, 09379) by following the manufacturer's instructions. Sample pellets were dissolved in DEPC-treated water and quantified using a spectrophotometer. RNA samples (1 μg) were heated at 65 °C for 5 min and subjected to reverse transcription (RT) using ReverTra Ace RT master mix (Toyobo) followed by incubation at 37 °C for 15 min, 50 °C for 5 min, and 98 °C for 5 min. cDNA was next amplified using gene-specific primers (Supplementary Table 1) together with KOD Fx (Toyobo, F0935K). PCR was performed under the following conditions: 94 °C for 2 min, and 35 cycles at 98 °C for 10 s, 55 °C for 30 s, and 68 °C for 40 s. Results were visualized on a 2% agarose gel. For real-time PCR, a master mix reagent was prepared with ReverTra Ace® qPCR RT kit (Toyobo) coupled with the real-time PCR system (Applied Biosystems).

### Thermal behavioral assay

Intercrossed wild-type and TMEM79⁻/⁻ littermates at 7-10 weeks of age were used for the thermal gradient ring (Ugo Basile) assay. The same background TRPV3⁻/⁻ mice were used as before[7]. All recordings were conducted between 11:00 to 20:00. The thermal apparatus was set at a temperature gradient from 10 to 45 °C (real floor temperature from 11.5 to 43.6 °C) by dividing it into two identical 12-increment temperature zones at 2.9 °C increments. All animals were acclimated for 30 minutes one day prior to the experiments on a device without a thermal gradient at an ambient temperature of 25 °C. All mice were released at the middle floor position, followed by unimpeded free movement for one h. Behaviors were individually tracked using a webcam (C920, Logitec) and transformed into processable information using ANY-maze software (Stoelting). The data were then analyzed and exported as "spending time" and "moving speed" from each specific zone using Microsoft Excel.

### Statistical analysis

All the data were obtained with at least three biological replicates. All statistical analysis was assessed with GraphPad Prism Version 9.1.2 or Origin.

### Reporting summary

Further information on research design is available in the Nature Portfolio Reporting Summary linked to this article.

## Data availability

All data and materials used in the analysis are available in the manuscript and Supplementary Information. Source data are provided in this paper.

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

## Acknowledgements

We thank Takuto Suito in NIPS for his contribution to the discussion. This work was supported by grants to MT from a Grant-in-Aid for Scientific Research from the Ministry of Education, Culture, Sports, Science, and Technology in Japan (#21H02667 and #20H05768).

## Author contributions

J.L. and M.T. designed the experiments and wrote the paper. J.L. conducted the experiments and analyzed data. R.Y. and M.K. acquired and interpreted the immunocytochemistry results. T.M. and M.A. provided the TMEM79 plasmid DNA and TMEM79$^{-/-}$ mice and joined the discussion.

## Competing interests

The authors declare no competing interests.
