## [Peer Review File · Nature Communications]

Involvement of skin TRPV3 in temperature detection regulated by TMEM79 in miceReviewer #1 (Remarks to the Author):

Comments to Author:

In this manuscript, Lei et al reported that TMEM79, a transmembrane protein, could regulate the functions of TRPV3. They found that heterologous expression of TMEM79 could suppress TRPV3-involved currents and it also modulated TRPV3 translocalization and degradation. They further demonstrated TMEM79 could mediate the electrophysiological properties of TRPV3 in vitro and affected animal behaviors in vivo. The authors proposed an interaction between TRPV3 and TMEM79 and their data suggested an underlying mechanism of TRPV3 in thermosensation in the skin.

Overall, the study is interesting and important. The experiments are well designed and conducted.

There are, however, in my point of view, some details should or need to be improved before it could be considered as a paper in NC:

1. The main shortcoming of this study is that similar results had been reported by previous studies (<https://pubmed.ncbi.nlm.nih.gov/15746429/>; <https://pubmed.ncbi.nlm.nih.gov/15175387/>), therefore, the novelty of this study is relatively limited.

2. Introduction has rather more descriptions on other functions of TRPV3 than its role in thermosensation.

3. It seems the results of Fig. 1e, f, g are contradictory to those of Supplementary Fig. 1a, b)?

4. The authors described that they performed whole-cell patch-clamp experiments only in calcium-free bath solutions, why avoid calcium?

5. Could the authors detail the method they used to heat the cells during the Ca²⁺ imaging assay (L547)?

6. 'data that provide direct support for the... underlying involvement of TRPV3 in thermosensation' (L36-37) --- this description is incomplete.

7. It seems that 'We previously reported that adenosine triphosphate (ATP) is used for the transmission of temperature information from skin keratinocytes to sensory neurons' (L49-50) --- this statement has been abruptly inserted or should be fully described to fit with its context.

8. 'This finding reveals a previously unreported modulation of TRPV3 ion channels with physiological significance to thermal sensation or skin' (L73-74) --- I think this paragraph should be 'This finding reveals a previously unreported modulation of TRPV3 ion channels with physiological significance to thermal sensation in the skin'?

9. In Discussion,

1) Some descriptions need to be revised for better logic and more readable (e.g. L276-277, L361-362).

2) Given the main topic of this study, the conclusion drawn at the end of the article seems to be too broad and too far.

Reviewer #2 (Remarks to the Author):

In this manuscript, Lei and coworkers provide evidence that an endoplasmic reticulum (ER) resident membrane protein TMEM79 is a negative regulator of the Ca²⁺-permeable cation channel TRPV3 in skin keratinocytes and this regulation somehow dampens the warm temperature preference of mice. TRPV3 is known to be involved in skin barrier formation and a number of skin diseases. It was first reported to be a temperature-regulated channel with sensitivity at the

innocuous warm temperatures. However, it remains debated whether TRPV3 contributes to temperature sensing in animals and whether the origin of the sensing resides in the keratinocytes or sensory nerves. Another issue is that despite the prevalent expression of TRPV3 mRNA in skin keratinocytes, it has been difficult to detect TRPV3-mediated currents and intracellular Ca²⁺ rise unless a combined use of 2APB and carvacrol, two very non-selective agonists with low potency, are used. The lack of highly selective TRPV3 agonist and antagonist only makes the situation worse in terms of interpreting the in vivo and in vitro data. Interestingly, the loss of function mutations of TMEM79 lead to many of the same skin illnesses as the gain-of-function mutations of TRPV3, such as atopic dermatitis, hair loss, and itch. Therefore, it makes sense to examine the functional and physical interaction between these two proteins. Here, the authors demonstrate that TMEM79 reduces TRPV3 currents activated by 2APB and heat both when heterologously expressed in HEK-293 cells and in primary skin keratinocytes, the two proteins physically interact with each other, and TMEM79 causes ER retention of TRPV3, leading to decreased cell surface expression and degradation of this channel. These findings reveal a new layer of TRPV3 regulation and provide further evidence about TRPV3 in temperature sensing. The work was well done and paper written clearly. The following comments should help improve the study.

- 1) The evidence for TRPV3 degradation in the presence of TMEM79 through the lysosome pathway is weak. First, the LAMP-1 positive puncta need to be labeled more clearly. This can be done by arrows or enlarging the area, highlighting the puncta colabeled by LAMP-1 and TRPV3. Second, without blocking proteasomes and lysosomes with specific inhibitors, one cannot conclude whether the degradation occurs through proteasomes or lysosomes simply based on the likely colocalization of the protein with lysosomes. Third, protein synthesis inhibitors should also be tested to rule out that the decrease was not due to reduced synthesis of TRPV3.
- 2) Ideally, the subcellular localization and interaction between TRPV3 and TMEM79 should be confirmed in native cells with the endogenous proteins. It is not true that "anti-TRPV3 antibodies are not available" (Lines 115-116). Quite a few companies sell TRPV3 antibodies. Have the authors tested any of them? Ref. 38 was cited in this paper to suggest ER localization of the TRPV3 mutants. If the authors trust their data, then the TRPV3 antibodies used in the cited reference must be good.
- 3) Would TMEM79 localize also to plasma membrane when coexpressed with TRPV3? If not, what was the evidence that ruled this out? Based on the images in Fig. 4a, TMEM79 is closely juxtaposed to the plasma membrane localized TRPV3. Was any of the PLA puncta (Fig. 4c) on the plasma membrane.
- 4) In Fig. 4b, the label for the western blots should be myc and flag instead of TRPV3 and TMEM79, respectively. The current labels give a wrong impression that the bands were detected by the TRPV3 and TMEM79 antibodies, which clearly is not the case here. If you want to indicate the nature of the proteins, you could label them as myc-TRPV3 and flag-TMEM79.
- 5) Lines 47-48 the sentence reads like "contribution contacts nerve endings...". This does not look right.
- 6) Line 74, "physiological significance" to which aspect(s) of "skin"?
- 7) Line 93, at this point, the evidence may suggest that "the differences in current amplitudes were LIKELY caused by differences in the expression level of mTRPV3 in the plasma membrane" but this is not proven until later biotinylation experiments.
- 8) Lines 99-100, "it becomes responsive to with a high-temperature threshold close to 50°C". This statement is wrong. The 50°C threshold refers to only the response to the first stimulation. With repeated heat stimulation, the threshold ranged from 31 to 39 °C, which is why TRPV3 is considered a warmth sensor, instead of a sensor for noxious heat.
- 9) Line 112, given that TMEM79 mainly affects cell surface expression of TRPV3, with the reduction on total TRPV3 protein levels being a minor change, I would suggest revising the subsection title to reflect the most prominent effect.
- 10) Line 121, "more predominant". Do you mean "more prominent"?
- 11) Lines 122-123, were these "TRPV3-evoked currents" or "TRPV3-mediated currents"?
- 12) Lines 123-124, are the current densities of myc-TRPV3 significantly lower than that of untagged TRPV3? Please comment on the difference.
- 13) Line 135, is there statistics on "This change was more apparent in cell populations than in single cells"?
- 14) Lines 136-137, please use arrows or enlarged images to show "vacuole-like structures" in the figure.
- 15) Line 153, do we know that "TMEM79 restricted TRPV3 translocation in the ER" or from the ER?

- 16) Line 214, if the difference is not statistically significant, they should not be referred to as "differed" no matter greatly or "only slightly".
- 17) Line 292, is there any evidence that TRPV3 proteins are improperly folded in the presence of TMEM79? And any evidence that the loss of TRPV3 on the plasma membrane was due to internalization rather than decreased forward trafficking? The cited study (ref. 38) does not provide convincing evidence that the G573 mutants traffic back to ER from the plasma membrane. They might just be retained in the ER. If they were retained in the ER, then they might bind better to TMEM79 than wild type TRPV3. However, the main problem with the cited study is that the examination might be performed on a bunch of unhealthy nearly dead cells. Therefore, it might be better to simply disregard any conclusion from this reference. As such, the speculation on the involvement of G573 and the S4-S5 linker in binding to TMEM79 (lines 307-308) does not withstand.
- 18) Line 317, maybe you want to say "distinct expression patterns" instead of "a unique expression pattern" here.
- 19) Line 397-398, how was mTMEM79 administered? Typically, administration refers to direct addition of the purified protein. If it is introduced through cotransfection, then it should be described as "transfection" or "overexpression".
- 20) Line 401, what super-resolution method was used for the imaging?
- 21) Lines 416-418, "Note that the highly co-transfected cells were myc-mTRPV3 and mTMEM79-flag in the cells". This sentence is hard to read.
- 22) Line 548, delete "AM" here. Most likely, your Fura-2 should have been de-esterified by this time. Only the de-esterified Fura-2 can report Ca²⁺ changes.

Reviewer #1 (Remarks to the Author):

1. The main shortcoming of this study is that similar results had been reported by previous studies (<https://pubmed.ncbi.nlm.nih.gov/15746429/>; <https://pubmed.ncbi.nlm.nih.gov/15175387/>), therefore, the novelty of this study is relatively limited.

Response:

As the reviewer pointed out, the involvement of TRPV3 in temperature detection was proposed before. However, no clear evidence for the concept has been provided due to the limitation of the temperature-dependent behavioral assay in mice. In our study, we substantiated the changes in temperature-dependent behaviors using a novel Thermal Gradient Ring in which we could evaluate both preference and avoidance, a much better method than a linear gradient apparatus with which Dr. Patapoutian reported less temperature sensitivity of TRPV3KO mice. Importantly, the involvement of TRPV3 function for detection of warm temperatures has not been reported before. Our behavioral data provide strong evidence to support the new concept that temperature can be detected by the skin in addition to sensory neurons through a thermosensitive TRPV3 channel. And more importantly, we provided a new regulation mechanism of TRPV3 through the interaction with TMEM79.

2. Introduction has rather more descriptions on other functions of TRPV3 than its role in thermosensation.

Response:

Thank you for this suggestion. We have reduced the description of other functions related to TRPV3, and we focused on its role in thermosensation.

3. It seems the results of Fig. 1e, f, g are contradictory to those of Supplementary Fig. 1a, b)?

Response:

The reviewer pointed out that they look contradictory. This was due to the fact that our legend for Suppl. Fig. 1 was not correct. It should have read, 'co-expression of TMEM79 does not affect the temperature thresholds for heat-evoked TRPV3 activation'. Importantly, co-expression of TMEM79 does affect the current amplitudes for the heat-evoked currents as shown in Fig. 1. Accordingly, we have corrected the title of Suppl. Fig. 1. We thank the reviewer for bringing this to our attention.

4. The authors described that they performed whole-cell patch-clamp experiments only

in calcium-free bath solutions, why avoid calcium?

Response:

TRPV3 is a unique ion channel because of its progressive sensitization upon repetitive stimulations. This property might be mediated by Ca^{2+} entering the cells as suggested in a previous paper (Xiao R, et al., 2008). Here, we observed the TRPV3 sensitization in Ca^{2+} -containing bath solution as shown in the figure below.

In addition, Ca^{2+} reportedly inhibits TRPV3 currents (Peier, A. M. et al., 2002; Xiao R, et al., 2008; Liu, B, et al., 2011). Accordingly, we decided to use a Ca^{2+} -free bath solution to evaluate TRPV3 currents to simplify activation because we could gather observations more quickly and currents were more stable compared with Ca^{2+} -containing bath solution.

5. Could the authors detail the method they used to heat the cells during the Ca^{2+} imaging assay (L547)?

Response:

Thank you for this suggestion. We have replaced the method as follows:

'The 300 μM 2-APB and carvacrol-containing or pre-heated bath solutions (140 mM NaCl, 5 mM KCl, 2 mM CaCl_2 , 2 mM MgCl_2 , 10 mM glucose and 10 mM HEPES, pH 7.4 adjusted with NaOH) were prepared before recording. The coverslip was first mounted in a chamber connected to a gravity flow system, allowing for delivery of different solutions, followed by perfusion of bath solution for about one minute. Subsequently, a 300 μM 2-APB/carvacrol cocktail or a heated bath solution was perfused, followed by 5 μM ionomycin (Sigma, I0634).'

6. ..'data that provide direct support for the underlying involvement of TRPV3 in thermosensation' (L36-37) --- this description is incomplete.

Response:

We thank the reviewer for this suggestion. We have corrected it accordingly.

7. It seems that 'We previously reported that adenosine triphosphate (ATP) is used for the transmission of temperature information from skin keratinocytes to sensory neurons' (L49-50) ---this statement has been abruptly inserted or should be fully described to fit with its context.

Response:

Thank you for this suggestion. We have rewritten this part in the revised manuscript.

8. 'This finding reveals a previously unreported modulation of TRPV3 ion channels with physiological significance to thermal sensation or skin' (L73-74) --- I think this paragraph should be 'This finding reveals a previously unreported modulation of TRPV3 ion channels with physiological significance to thermal sensation in the skin'?

Response:

We thank the reviewer for this suggestion. Together with reviewer #2's comment 6, we have corrected it with the following: 'This finding reveals a previously unreported modulation of TRPV3 ion channels, and indicates that skin keratinocytes play a role in conveying thermal information.'

9. In Discussion,

1) Some descriptions need to be revised for better logic and more readable (e.g. L276-277, L361-362).

Response:

This was a helpful suggestion. We have rewritten this part in the revised manuscript.

2) Given the main topic of this study, the conclusion drew at the end of the article seems to be too broad and too far.

Response:

Based upon the reviewer's suggestion, we have shortened the Discussion by focusing on the thermosensing ability of TRPV3 and related issues. In addition, we have described the itching phenotype since it was observed in TMEM79KO mice and could be partly explained by the increased TRPV3 in the skin. Moreover, we have added some perspectives regarding unanswered questions that can be addressed in future studies.

Reviewer #2 (Remarks to the Author):

1. The evidence for TRPV3 degradation in the presence of TMEM79 through the

lysosome pathway is weak. First, the LAMP-1 positive puncta need to be labeled more clearly. This can be done by arrows or enlarging the area, highlighting the puncta colabeled by LAMP-1 and TRPV3. Second, without blocking proteasomes and lysosomes with specific inhibitors, one cannot conclude whether the degradation occurs through proteasomes or lysosomes simply based on the likely colocalization of the protein with lysosomes. Third, protein synthesis inhibitors should also be tested to rule out that the decrease was not due to reduced synthesis of TRPV3.

Responses:

We thank the reviewer for these comments and suggestions.

First, we highlighted the puncta more clearly in the revised Figure 3b.

Second, based upon the suggestion by the reviewer, we performed experiments with 2 lysosome inhibitors (chloroquine and bafilomycin A) and 2 proteasome inhibitors (MG132 and Lactacystin), and we examined the amounts of myc-tagged TRPV3 by Western blotting. The amounts of TRPV3 protein were reduced in the control as shown in the previous Figure 2 whereas such a reduction was not observed in the cells treated with either lysosome inhibitors. On the other hand, changes in the TRPV3 amounts were reduced in the cells treated with either proteasome inhibitors. These results indicate that degradation of TRPV3 proteins predominantly occurred in the lysosome pathway rather than the proteasome one. We have included these data in the revised Figure 3.

Third, we treated the cells with a protein synthesis inhibitor, cycloheximide (CHX, 100 $\mu\text{g}/\text{ml}$) 24 h after transfection. We examined the amount of myc-tagged TRPV3 protein at 0, 3, 6, 12, and 24 h after starting the CHX treatment in cells with and without co-expression of TMEM79. We found that the normalized amount of TRPV3 was significantly smaller at 24 h CHX treatment in the cells co-expressing TMEM79 than in cells expressing TRPV3 alone. This result suggests that the reduction of TRPV3 is not caused by the reduced protein synthesis although it is hard to observe the effect of protein synthesis inhibition in overexpression systems like the one used in our HEK293T cells. Rather, the apparently greater reduction of TRPV3 in the experiment could be caused by the increased degradation occurring in the lysosomal pathway as observed above. Accordingly, we have added these data in the revised Figure 3c-f.

2. Ideally, the subcellular localization and interaction between TRPV3 and TMEM79 should be confirmed in native cells with the endogenous proteins. It is not true that “anti-TRPV3 antibodies are not available” (Lines 115-116). Quite a few companies sell TRPV3 antibodies. Have the authors tested any of them? Ref. 38 was cited in this paper to

suggest ER localization of the TRPV3 mutants. If the authors trust their data, then the TRPV3 antibodies used in the cited reference must be good.

Response:

We agree that we should evaluate the subcellular localization and interaction between TRPV3 and TMEM79 in native cells with endogenous proteins. We examined the 5 commercially available anti-TRPV3 antibodies including the one used in ref. 38 (from Alomone) in the Western blot analysis with skin keratinocytes, not in HEK293T cells exogenously expressing TRPV3. Unfortunately, however, no clear bands at the expected molecular weights of TRPV3 were detected by the 4 antibodies as shown below.

One antibody (from Novus) looked useful showing a clear band that was not observed in TRPV3KO keratinocytes. However, when we examined the IP samples, we could not detect the TRPV3-specific band as shown below.

Accordingly, we concluded that we cannot detect the TRPV3 proteins in the native cells. Nonetheless, we believe that our experiments with myc-TRPV3 and TMEM79-flag clearly show the subcellular localization and interaction between TRPV3 and TMEM79 in HEK293T cells, and patch-clamp data with mouse skin keratinocytes support the results.

These data indicate that the results shown in ref. 38 might not be correct. Accordingly, we decided not to cite the work.

3. Would TMEM79 localize also to plasma membrane when coexpressed with TRPV3? If not, what was the evidence that ruled this out? Based on the images in Fig. 4a, TMEM79 is closely juxtaposed to the plasma membrane localized TRPV3. Was any of the PLA puncta (Fig. 4c) on the plasma membrane.

Response:

As shown in the original Figure 2a, some TMEM79 proteins are in the plasma membrane. This result is consistent with the apparent plasma membrane expression of TMEM79 as shown in Figure 4a. As suggested by the reviewer, we have further examined the precise localization of the plasma membrane marker Na⁺/K⁺-ATPase and the PLA signals by super-resolution imaging using an LSM800 Airyscan. The super-resolution imaging enables a resolution of 140 nm in the XY plane and 200-350nm in the XZ or YZ planes. As shown in the revised Figure 4c and d, most PLA signals were clearly localized on the immunoreactivity of Na⁺/K⁺-ATPase. Because the original Figure 4c consisted of non-confocal images, the apical membrane signals were imposed on the basal signals, and the PLA signals seemed to be distributed to the cytoplasm. The graph of fluorescent intensity profile in the revised Figure 4d shows that PLA signals were on the Na⁺/K⁺-ATPase in the cells expressing TRPV3 and TMEM79, suggesting that the interaction was on or adjacent to the plasma membrane. Accordingly, we have replaced the previous Figure 4c with a new one (Figures 4c and d) showing the overlap between PLA puncta and membrane protein marker.

4. In Fig. 4b, the label for the western blots should be myc and flag instead of TRPV3 and TMEM79, respectively. The current labels give a wrong impression that the bands were detected by the TRPV3 and TMEM79 antibodies, which clearly is not the case here. If you want to indicate the nature of the proteins, you could label them as myc-TRPV3

and flag-TMEM79.

Response:

We thank the reviewer for this suggestion. We have corrected it accordingly.

5. Lines 47-48 the sentence reads like “contribution contacts nerve endings....”. This does not look right.

Response:

We thank the reviewer for this suggestion. We have corrected it accordingly.

6) Line 74, “physiological significance” to which aspect(s) of “skin”?

Response:

We thank the reviewer. We have revised the sentence as follows: ‘This finding reveals a previously unreported modulation of TRPV3 ion channels, and indicates that skin keratinocytes play a role in conveying thermal information.’

7) Line 93, at this point, the evidence may suggest that “the differences in current amplitudes were LIKELY caused by differences in the expression level of mTRPV3 in the plasma membrane” but this is not proven until later biotinylation experiments.

Response:

We thank the reviewer for this suggestion. We have added the word ‘likely’ as suggested.

8) Lines 99-100, “it becomes responsive to with a high-temperature threshold close to 50°C”. This statement is wrong. The 50°C threshold refers to only the response to the first stimulation. With repeated heat stimulation, the threshold ranged from 31 to 39 °C, which is why TRPV3 is considered a warmth sensor, instead of a sensor for noxious heat.

Response:

We appreciate the reviewer’s suggestion. We have added the word ‘initial’ in the revised manuscript.

9) Line 112, given that TMEM79 mainly affects cell surface expression of TRPV3, with the reduction on total TRPV3 protein levels being a minor change, I would suggest revising the subsection title to reflect the most prominent effect.

Response:

Thank you for this suggestion. We have changed the title to 'TMEM79 downregulates TRPV3 plasma membrane protein levels'.

10) Line 121, "more predominant". Do you mean "more prominent"?

Response:

We thank the reviewer's suggestion. We have corrected it accordingly.

11) Lines 122-123, were these "TRPV3-evoked currents" or "TRPV3-mediated currents"?

Response:

Thank you. We have replaced "TRPV3-evoked currents" with "TRPV3-mediated currents".

12) Lines 123-124, are the current densities of myc-TRPV3 significantly lower than that of untagged TRPV3? Please comment on the difference.

Response:

We appreciate this comment. All the current amplitude data were described in the previous manuscript. The myc-tagged TRPV3-mediated currents were significantly smaller than untagged TRPV3-mediated currents regardless of the presence of TMEM79. However, TRPV3 currents were always smaller when TMEM79 was co-expressed. Accordingly, we have added the following sentence when describing the myc-tagged TRPV3-mediated current responses, 'while myc-tagged TRPV3-mediated currents were significantly smaller than untagged TRPV3-mediated currents.'

13) Line 135, is there statistics on "This change was more apparent in cell populations than in single cells"?

Response:

We deleted this description because it is difficult to quantify this phenomenon.

14) Lines 136-137, please use arrows or enlarged images to show "vacuole-like structures" in the figure.

Response:

In accord with the suggestion, we have enlarged images to show "vacuole-like structures" with arrows in Figure 2c.

15) Line 153, do we know that “TMEM79 restricted TRPV3 translocation in the ER” or from the ER?

Response:

We do not know whether TMEM79 restricted TRPV3 translocation in the ER or from the ER. Accordingly, we have removed the description, and used the sentence, ‘TMEM79 caused accumulation of TRPV3 in the ER.’

16) Line 214, if the difference is not statistically significant, they should not be referred to as “differed” no matter greatly or “only slightly”.

Response:

Thank you for this remark. We have revised the description as follows: ‘whereas GSK101-induced currents did not differ between them’.

17) Line 292, is there any evidence that TRPV3 proteins are improperly folded in the presence of TMEM79? And any evidence that the loss of TRPV3 on the plasma membrane was due to internalization rather than decreased forward trafficking? The cited study (ref. 38) does not provide convincing evidence that the G573 mutants traffic back to ER from the plasma membrane. They might just be retained in the ER. If they were retained in the ER, then they might bind better to TMEM79 than wild type TRPV3. However, the main problem with the cited study is that the examination might be performed on a bunch of unhealthy nearly dead cells. Therefore, it might be better to simply disregard any conclusion from this reference. As such, the speculation on the involvement of G573 and the S4-S5 linker in binding to TMEM79 (lines 307-308) does not withstand.

Response:

We do not have any data regarding protein folding and internalization although we would like to define the processes that reduce the plasma membrane expression of TRPV3 by TMEM79. As described above, we concluded that the anti-TRPV3 antibody (from Novus) cannot be used for protein analysis and decided not to cite the work in ref. 38. The folding and internalization of TRPV3 are beyond the current project, and we chose not to discuss it by citing the data in ref. 38.

18) Line 317, maybe you want to say “distinct expression patterns” instead of “a unique expression pattern” here.

Response:

We thank the reviewer for this suggestion. We have corrected it accordingly.

19) Line 397-398, how was mTMEM79 administered? Typically, administration refers to direct addition of the purified protein. If it is introduced through cotransfection, then it should be described as “transfection” or “overexpression”.

Response:

We thank the reviewer’s suggestion. We have used the term ‘overexpression’.

20) Line 401, what super-resolution method was used for the imaging?

Response:

We thank the reviewer’s comment. We acquired images with Zeiss LSM800 or LSM880 laser scanning microscope equipped with an Airyscan detector module, which enhanced lateral and axial resolution (Huff et al., 2015; Bond et al., 2022). Airyscan detectors consist of 32 small point detectors (each detector element behaving as a 0.2 airy unit pinhole) that collectively maintain the light collection efficiency of a 1.25 airy unit pinhole, improving the signal-to-noise ratio without compromising resolution. The images were further processed for Airyscan processing (a linear deconvolution step for each of the 32 individual elements followed by pixel reassignment of each element back to the center position). We realized the resolution of 140 nm at XY plane 200-350 nm at XZ or YZ planes. Accordingly, we added a description about the super-resolution imaging method in the “Immunofluorescence” section of the method part.

21) Lines 416-418, “Note that the highly co-transfected cells were myc-mTRPV3 and mTMEM79-flag in the cells”. This sentence is hard to read.

Response:

Thank you. We decided to remove this sentence because it is not informative based on the representative images.

22) Line 548, delete “AM” here. Most likely, your Fura-2 should have been de-esterified by this time. Only the de-esterified Fura-2 can report Ca²⁺ changes.

Response:

We thank the reviewer’s suggestion. We have corrected it accordingly.

Reviewer #1 (Remarks to the Author):

Remarks to the Author:

The Authors have addressed most of my previous concerns and I find the manuscript significantly improved. Nonetheless, I still take issue with the statements which the Authors made as a response to my first comment.

As we can learn from previous reports of TRPV3, there are studies using mouse models to examine temperature related behaviors and some even concluded contradictory to the results of the current study. The Authors mentioned that 'the involvement of TRPV3 function for detection of warm temperatures has not been reported before'---seems the Authors have overlooked some articles/review such as: <https://pubmed.ncbi.nlm.nih.gov/31636415/>;
<https://pubmed.ncbi.nlm.nih.gov/24925072/>;
<https://www.sciencedirect.com/science/article/pii/S0143416003000630?via=ihub>).

Therefore, I suggest that the Authors to fully address these aspects by comparing with more prior data and further revise the manuscript so as to correctly highlight their contributions to the field.

Minor:

In Abstract, 'These finding provide direct support for the underlying involvement of TRPV3 in thermosensation' (L36), which should read as: These findings...

Reviewer #2 (Remarks to the Author):

The revised ms addressed all questions that I raised. There are just two minor language issues:

- 1) Line 54-55, the sentence reads like that TRPV3 and TRPV4 had abnormal behavior. This is incorrect. Do you mean that the TRPV3 and TRPV4 knockout mice have abnormal behaviors in temperature gradient assays?
- 2) Line 117-118, it is not clear to readers what you mean by "anti-TRPV3 antibodies are hardly available". Do you mean that anti-TRPV3 antibodies with suitable quality are not available?

REVIEWERS' COMMENTS

Reviewer #1 (Remarks to the Author):

The Authors have addressed most of my previous concerns and I find the manuscript significantly improved. Nonetheless, I still take issue with the statements which the Authors made as a response to my first comment.

As we can learn from previous reports of TRPV3, there are studies using mouse models to examine temperature related behaviors and some even concluded contradictory to the results of the current study. The Authors mentioned that 'the involvement of TRPV3 function for detection of warm temperatures has not been reported before'---seems the Authors have overlooked some articles/review such as: <https://pubmed.ncbi.nlm.nih.gov/31636415/>; <https://pubmed.ncbi.nlm.nih.gov/24925072/>; <https://www.sciencedirect.com/science/article/pii/S0143416003000630?via=ihub>).

Therefore, I suggest that the Authors to fully address these aspects by comparing with more prior data and further revise the manuscript so as to correctly highlight their contributions to the field.

Response:

We thank the reviewer for the comments, which helped us to improve our manuscript. We have added the description about others' studies (L304-306 and L357-377) in the discussion section including the three papers the reviewer suggested. The general difference between the previous and our studies was mainly addressed. We hope this does not overlook the significance of others' contributions.

Minor:

In Abstract, 'These finding provide direct support for the underlying involvement of TRPV3 in thermosensation' (L36), which should read as: These findings...

Response:

We thank the reviewer for this suggestion. We have corrected it accordingly.

Reviewer #2 (Remarks to the Author):

The revised ms addressed all questions that I raised. There are just two minor language issues:

1)Line 54-55, the sentence reads like that TRPV3 and TRPV4 had abnormal behavior. This is incorrect. Do you mean that the TRPV3 and TRPV4 knockout mice have abnormal behaviors in temperature gradient assays?

Response:

We thank the reviewer for pointing this out. We are sorry to make an ambiguous sentence. Accordingly, we have corrected it with the following: 'it has been reported that TRPV3 and TRPV4 are important for warmth sensation because of their warmth-induced activation and prevalent expression in keratinocytes, and because of abnormal behavior of mutant mice in temperature gradient assays'.

2) Line 117-118, it is not clear to readers what you mean by "anti-TRPV3 antibodies are hardly available". Do you mean that anti-TRPV3 antibodies with suitable quality are not available?

Response:

As the reviewer said the quality of commercial anti-TRPV3 antibodies is not effective in our hands. So, we have corrected the sentence to 'Given the fact that reliable anti-TRPV3 antibodies are hardly available'.